# ACTIONS AS LANGUAGE: FINE-TUNING VLMS INTO VLAS WITHOUT CATASTROPHIC FORGETTING

**Asher J. Hancock,**[*] **Xindi Wu, Lihan Zha, Olga Russakovsky, Anirudha Majumdar**
Princeton University

## ABSTRACT

Fine-tuning vision-language models (VLMs) on robot teleoperation data to create vision-language-action (VLA) models is a promising paradigm for training generalist policies, but it suffers from a fundamental tradeoff: learning to produce actions often diminishes the VLM's foundational reasoning and multimodal understanding, hindering generalization to novel scenarios, instruction following, and semantic understanding. We argue that this catastrophic forgetting is due to a distribution mismatch between the VLM's internet-scale pretraining corpus and the robotics fine-tuning data. Inspired by this observation, we introduce VLM2VLA: a VLA training paradigm that first resolves this mismatch at the data level by *representing low-level actions with natural language*. This alignment makes it possible to train VLAs *solely with Low-Rank Adaptation (LoRA)*, thereby minimally modifying the VLM backbone and averting catastrophic forgetting. As a result, the VLM can be fine-tuned on robot teleoperation data without fundamentally altering the underlying architecture and without expensive co-training on internet-scale VLM datasets. Through extensive Visual Question Answering (VQA) studies and over 800 real-world robotics experiments, we demonstrate that VLM2VLA preserves the VLM's core capabilities, enabling zero-shot generalization to novel tasks that require open-world semantic reasoning and multilingual instruction following.

## 1 INTRODUCTION

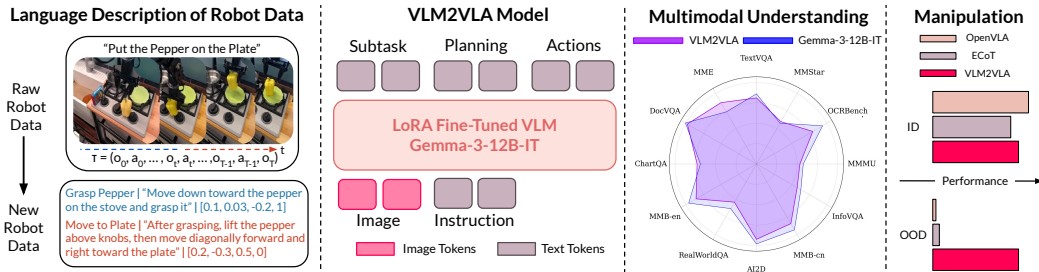

Figure 1: We present VLM2VLA, a data pipeline and training methodology for fine-tuning VLMs into VLAs while preserving their foundational perceptual and reasoning capabilities. Our policy retains its pretraining knowledge, enabling strong VQA performance and superior generalization in real robotic manipulation tasks.

The pursuit of generalist robot policies capable of understanding and executing human commands has been significantly advanced by the integration of vision-language models (VLMs) (Team et al., 2025b; Bai et al., 2025; Team, 2025) throughout the autonomy stack. Trained on internet-scale datasets of image-text pairs, these models have acquired sophisticated capabilities in perception, semantic understanding, and commonsense reasoning. To endow robots with similar capabilities, the prevailing paradigm involves fine-tuning pretrained VLMs on robot demonstration data, transforming them into vision-language-action models (VLAs) that map from natural language commands and visual observations to robot actions. This approach has yielded impressive results across a wide range

---

[*]Correspondence to: ajhancock@princeton.edu

of robotic manipulation tasks (Kim et al., 2024; Black et al., 2024; Intelligence et al., 2025; Gao et al., 2025; Brohan et al., 2023b;a; Padalkar et al., 2024; Team et al., 2025a).

However, the standard methodology of fundamentally modifying the VLM's architecture, tokenization vocabulary, or a combination thereof, coupled with full parameter fine-tuning on robot imitation learning data, introduces a crucial yet often overlooked trade-off. In adapting the VLM for robotic control, we risk overfitting to the robot fine-tuning data, thereby overwriting the general-purpose world knowledge acquired during pretraining (see Fig. 2). The consequences of this trade-off are far-reaching: current VLAs often exhibit a diminished ability to generalize to novel objects, handle linguistic variations, be robust to distractions, or reason about concepts outside the narrow scope of their robotic training data (Kim et al., 2024; Brohan et al., 2023a; Hancock et al., 2024; Zhou et al., 2025b).

Preserving the VLM's foundational world knowledge during VLA fine-tuning is essential for creating truly generalist robot policies; consequently, numerous techniques have been developed to address this challenge. The most common approach is to co-train with non-robotic data. This training regimen regularizes the VLM against overfitting to robot datasets, thereby mitigating the loss of its foundational capabilities (Brohan et al., 2023a; Intelligence et al., 2025; Team et al., 2025a).

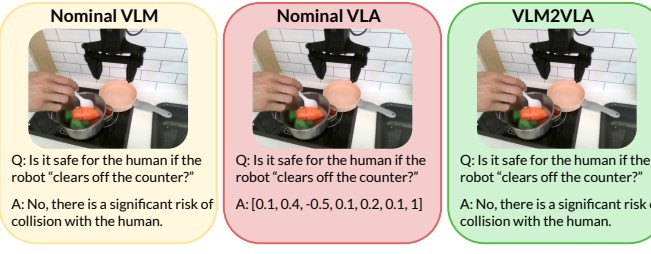

Figure 2: Traditional VLA training procedures often overfit to the robot training data, sacrificing their original reasoning capabilities for low-level action prediction (center). In contrast, VLM2VLA (right) preserves the world understanding of the nominal VLM (left), allowing the model to reason about potential safety risks instead of just motor commands.

2025a). While these methods can mitigate knowledge loss, co-training with VLM-scale datasets is inherently expensive and requires a carefully constructed dataset mixture for optimal performance.

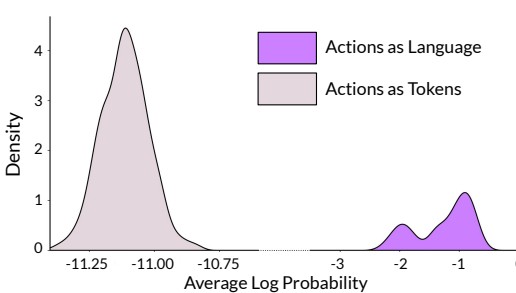

Figure 3: Distribution of action probabilities under Gemma-3-12B-IT *before* fine-tuning on robot teleoperation data. The model assigns significantly higher log-probabilities to actions represented as language compared to those defined by explicit tokenization modifications, e.g., least likely token assignment.

This paper aims to preserve the world knowledge of the VLM while adapting it for robotic control *without* co-training. We address this issue by resolving the *distribution mismatch between the low-level action spaces needed for robotic control and the image-text distributions of the VLM's pretraining corpus*. This mismatch often compels researchers to use full parameter fine-tuning when training VLAs, which contributes to catastrophic forgetting by overfitting to the robot teleoperation data.

Our key insight is that while parameter-efficient methods like Low-Rank Adaptation (LoRA) (Hu et al., 2021) can avert catastrophic forgetting, their effectiveness relies on the fine-tuning data being sufficiently close to the model's pretrained representations. We therefore propose resolving this representational mismatch *at the data level*. Our data-centric approach *re-represents robot actions as natural language descriptions, thereby aligning the VLA fine-tuning data directly with the VLM's pretrained representation space*. This alignment enables LoRA to effectively adapt the VLM for robotic control without significantly perturbing its pretrained weights. Fig. 3 illustrates this idea, showing our language-based actions are assigned significantly higher probabilities by the VLM backbone than actions mapped to arbitrary tokens, a common strategy in state-of-the-art VLAs (Brohan et al., 2023b; Kim et al., 2024; Brohan et al., 2023a; Padalkar et al., 2024; Lee et al., 2025). Our method is model agnostic and simple to implement, obviating the need for sophisticated architectures, complex co-training schemes, or multi-stage training procedures to achieve robust knowledge retention and superior generalization capabilities.

**Statement of Contributions.** We present VLM2VLA, a data pipeline and training methodology for fine-tuning VLMs into VLAs while preserving their foundational perceptual and reasoning capabilities. Our core contributions are as follows: **1) Representing actions as language**: We propose translating low-level robotic imitation data into text, thereby aligning the VLA fine-tuning data with the VLM's pretraining distribution to mitigate catastrophic forgetting. **2) A data re-labeling and training pipeline for knowledge retention**: Building on our action representation, we present a scalable methodology for re-labeling robot teleoperation datasets for fine-tuning a VLM into a VLA through LoRA. **3) Empirical validation of action and reasoning capabilities**: We provide extensive empirical validation showing our VLA preserves a suite of crucial capabilities that are often lost in other state-of-the-art models. Specifically, our method's efficacy is demonstrated through extensive real-world evaluation (over 800 robotic experiments), showing generalization to novel tasks with objects and language instructions unseen during training. Moreover, our policy averts catastrophic forgetting, retaining over 85% of the base model's performance across challenging VQA benchmarks.

## 2   RELATED WORK

**Vision-Language Action (VLA) Models**: VLA models, which fine-tune pretrained VLM backbones for robotic control, have become the dominant paradigm in training generalist robotic policies (Kim et al., 2024; Black et al., 2024; Qu et al., 2025; Li et al., 2024; Driess et al., 2023; Team et al., 2025a; NVIDIA et al., 2025; Yang et al., 2025b; Lee et al., 2025). The central promise of this approach is knowledge transfer: one can translate the rich semantic knowledge of the VLM, learned from internet-scale pretraining, directly to action prediction. One challenge in adapting transformer-based VLMs for action prediction is bridging the gap between their discrete, token-based nature and the continuous, vector-valued representations utilized in robotic control. Consequently, many state-of-the-art VLAs diverge architecturally in how they address the problem of action generation.

**Action Representation in VLAs**: The first dominant strategy is discretization, where continuous action vectors are explicitly mapped to a finite set of tokens (Lee et al., 2025; Pertsch et al., 2025), typically the VLM's least likely tokens (Brohan et al., 2023b;a; Padalkar et al., 2024; Kim et al., 2024). This approach casts robot control as a standard next-token prediction problem, allowing the VLA to generate outputs autoregressively. The second paradigmatic strategy augments the VLM with a separate, lightweight action head (e.g., using diffusion or flow-matching) to generate continuous actions directly (Team et al., 2024; Black et al., 2024; Intelligence et al., 2025; Wen et al., 2025; Zhou et al., 2025b;a). While this design choice enables faster action prediction (Pertsch et al., 2025), it introduces new, randomly initialized parameters to the VLM during VLA fine-tuning which can corrupt the original pretrained representations (Driess et al., 2025; Zhou et al., 2025b;a).

Our work introduces a third approach: *representing actions as natural language descriptions all within the VLM's existing vocabulary*. For instance, the command 'move forward by 4.2 centimeters' is treated as a standard text string, allowing our method to bootstrap the VLM's intrinsic understanding of numerical magnitude for grounding in physical space. We provide extensive experimental evidence in Section 4.2 that this 'actions as language' representation is more effective for policy learning via LoRA fine-tuning than strategies based on least likely token assignment. Unlike the concurrent work of Grover et al. (2025), which use full parameter fine-tuning and co-training, we use only LoRA and our action representation to preserve the VLM's foundational reasoning capabilities.

**Reasoning Models**: To improve performance on complex, long-horizon tasks, a growing body of work has focused on enabling VLAs to perform explicit reasoning. These methods typically use a chain-of-thought approach (Wei et al., 2023), where the model first generates a high-level plan or reasoning trace that then conditions the prediction of low-level actions. The form of this reasoning varies across the literature. One set of approaches provides task-level context, leveraging (vision) language models to generate natural language sub-goals or task plans that guide the policy (Intelligence et al., 2025; Lin et al., 2025; Shi et al., 2025; Zhao et al., 2025; Ahn et al., 2022; Silver et al., 2023; Shi et al., 2024; Clark et al., 2025; Huang et al., 2022; Zeng et al., 2022). A second set of works condition the policy on more structured, mid-level representations like trajectory sketches, code, or spatial affordances (Belkhale et al., 2024; Nasiriany et al., 2024; Li et al., 2025; Liang et al., 2023; Cheng et al., 2025; Zhi et al., 2025; Zawalski et al., 2025).

Figure 4: VLM2VLA's pipeline for annotating existing robot datasets $\mathcal{D}_{\text{rob}}$ into $\mathcal{D}_{\text{lan}}$ described via natural language. We use Gemini 2.5 (Team, 2025) to decompose each trajectory into sub-trajectories, each with an associated subtask, motion plan, and action chunk.

Our work is distinct from prior VLAs and embodied reasoning models in that (1) we not only represent high-level reasoning in language, but also *represent low-level actions (e.g., end-effector movements) using text*, (2) we do not utilize any explicit action decoders, and (3) *we train the VLA solely with LoRA*. We posit that modifications to the VLM, combined with full-parameter fine-tuning, create a distribution shift deleterious to the VLM's world knowledge and reasoning capabilities. We demonstrate evidence of this claim in Section 4.1.

**Co-Training for Knowledge Retention**: A critical and often overlooked phenomenon of many VLAs is catastrophic forgetting, where the rich, general-purpose knowledge of the base VLM is lost (Zhou et al., 2025b). To combat this, the most common effective strategy has been large-scale co-training (Brohan et al., 2023a; Intelligence et al., 2025; Zhou et al., 2025a; Lee et al., 2025), which mixes robotic and non-robotic datasets (e.g., VQA, image captioning). Co-training effectively regularizes the VLA fine-tuning process, whereby the VLM is continuously trained on data drawn from its pretraining distribution to preserve general reasoning and visual understanding. However, co-training requires careful tuning of the mixture ratio between robotic and web data. This hyperparameter is non-trivial and expensive to specify, requiring numerous training runs and real-world evaluations to find an effective setting for both VQA performance and dexterous robotic control.

**Advanced Training Schemes to Avert Catastrophic Forgetting**: Sophisticated VLA training procedures in Mixture-of-Experts (MoE)-style VLAs have recently been developed as an alternative means of preventing catastrophic forgetting. For instance, Driess et al. (2025) attempt to shield the VLM from otherwise destructive gradient updates of the action module via stop-gradient operators. Moreover, Zhou et al. (2025a) have explored a sequential training procedure whereby the VLM's weights are frozen during training of the action expert. Our work challenges the necessity of co-training and other sophisticated training schemes to prevent catastrophic forgetting; by representing robot actions as natural language, our VLA can be effectively fine-tuned with only low-rank adaptation, a simple procedure requiring no major architectural changes.

## 3  METHODOLOGY

In this section, we discuss our VLA training pipeline. Our approach is predicated on a simple principle: to successfully leverage LoRA fine-tuning and avert catastrophic forgetting, we must first align the robot data with the VLM's existing representational space. To this end, we begin by first translating robot teleoperation datasets into natural language descriptions. We posit that this data transformation minimizes distribution shift incurred during LoRA fine-tuning (as evidence by Fig. 3), thereby retaining the VLM's nominal capabilities while enabling effective robot control.

### 3.1  ACTIONS AS LANGUAGE: A HIERARCHICAL REPRESENTATION

VLM2VLA frames action prediction as a three-stage VQA hierarchical reasoning process, which has been shown to improve generalization to new tasks and environments (Zawalski et al., 2025; Intelligence et al., 2025); the fundamental difference between our action representation and other VLAs (Belkhale et al., 2024; Kim et al., 2024; Zawalski et al., 2025; Brohan et al., 2023a; Padalkar et al., 2024; Clark et al., 2025) is that high-level and low-level actions are represented directly via language. Let the main task be broken into a series of $N$ steps, indexed by $i$:

**High-Level Subtask Prediction** ($l_i$): given observation $\bar{o}_i$ and language instruction $L$, the model first describes what immediate subtask $l_i$ is necessary to complete the main task.

**Mid-Level Motion Planning** ($m_i$): conditioned on the current subtask and observation, the model generates a spatially informative motion plan $m_i$, with respect to the robot's end-effector, detailing the movements necessary to complete the subtask. Our motion plans describe only directional movements, e.g., 'move left' and 'move down and slightly forward.' This coarse description was specifically chosen to benefit from the VLM's latent spatial reasoning, which excels at these types of tasks (Team et al., 2025b; Bai et al., 2025).

**Low-Level Action Generation** ($\bar{a}_i$): conditioned on the current subtask and motion plan, the policy generates a variable-length action-chunk $\bar{a}_i$ to execute directly on the robot. In practice, the action-chunk is a list of lists, where each inner list contains the individual commands *represented as text* for each degree of freedom (DoF) of the robot. In this work, we only considered translational DoFs.

Formally, the distribution we seek to capture with our VLA is $p_\theta(\bar{a}_i, m_i, l_i | \bar{o}_i, L)$, which we decompose as:

$$p_\theta(\bar{a}_i, m_i, l_i | \bar{o}_i, L) = \underbrace{p_\theta(l_i | \bar{o}_i, L)}_{\substack{\text{1) Subtask} \\ \text{Prediction}}} \underbrace{p_\theta(m_i | l_i, \bar{o}_i)}_{\substack{\text{2) Motion} \\ \text{Planning}}} \underbrace{p_\theta(\bar{a}_i | m_i, l_i, \bar{o}_i)}_{\substack{\text{3) Action} \\ \text{Generation}}}, \quad (1)$$

where parameters $\theta$ are the weights of the robot policy. This parameterization corresponds to a vision-language model (transformer) taking as input a sequence of multimodal image-text input tokens and predicting a sequence of output text tokens. All observations reside in RGB image space.

The decomposition given in Equation (1) describes how our VLA operates at test-time. Every action-chunk prediction is conditioned on the current observation, subtask description, and proposed motion-plan; in practice, we generate all $N$ subtasks at once given the initial observation $\bar{o}_0$ and keep this set fixed for the duration of the rollout.

To improve robustness of the policy, we operate in closed-loop fashion with a verifier. At the end of every action-generation cycle, the verifier determines if the model should re-try the current subtask or proceed onto the next one, a process which continues until all $N$ subtasks are completed. In this work, we utilize Gemini 2.5 Pro (Team, 2025) as the verifier (with details in Appendix D.5).

## 3.2 Data Curation: Translating Robot Trajectories into a Hierarchical Format

To teach a VLM the aforementioned spatially-grounded reasoning chain, we must first re-label existing robot datasets in natural language. We assume access to a dataset of human teleoperated robot trajectories, $\mathcal{D}_{\text{rob}}$, where each trajectory is a state-action sequence $\tau = \{(o_t, a_t)\}_{t=0}^T$ of length $T$ controlling the relative position of the robot's end-effector (Padalkar et al., 2024; Khazatsky et al., 2025; Walke et al., 2024). Each trajectory comes equipped with a main-task language instruction $L$. Our goal is to automatically describe each trajectory $\tau \in \mathcal{D}_{\text{rob}}$ at the levels described in Section 3.1 to construct a new robot dataset annotated with natural language, $\mathcal{D}_{\text{lan}}$. Herein, we utilize the subset of the Bridgev2 dataset equipped with main-task instructions as $\mathcal{D}_{\text{rob}}$ (Walke et al., 2024).

Our pipeline for constructing $\mathcal{D}_{\text{lan}}$ is similar in spirit to the data curation strategies of Clark et al. (2025); Zawalski et al. (2025). We use Gemini (Team, 2025) to automatically annotate trajectories and construct our our natural language dataset (see Fig. 4 for an illustration). Specifically, Gemini will decompose each trajectory into a sequence of $N$ steps. Each step $i$ comes equipped with an initial observation $\bar{o}_i$, subtask $l_i$, motion plan $m_i$, and variable-length action chunk $\bar{a}_i$, allowing us to represent the trajectory described via natural language, $\bar{\tau}$, as the tuple $\bar{\tau} = \{(\bar{o}_i, l_i, m_i, \bar{a}_i)\}_{i=0}^{N-1} \in \mathcal{D}_{\text{lan}}$. See Appendix B.1 and B.2 for additional details on dataset curation.

In summary, we transform our original robot dataset of state-action pairs into a dataset of image-text pairs, thereby casting robotic control as a standard supervised fine-tuning task. We fine-tune the Gemma-3-12B-IT model (Team et al., 2025b) using LoRA applied to all its linear modules using the cross-entropy loss. This approach instills action prediction capabilities while preserving the VLM's world knowledge. Additional training details are outlined in Appendix E.

## 4 Experiments

In this section, we evaluate VLM2VLA on a suite of tasks to answer the following questions:

Table 1: **Multimodal understanding evaluation.** Comparison of VLMs and VLAs across multimodal understanding benchmarks. We compare against Prismatic VLM (Karamcheti et al., 2024), OpenVLA (Kim et al., 2024), ECoT (Zawalski et al., 2025), Gemma-3 (Team et al., 2025b), Molmo Deitke et al. (2024), MolmoAct (Lee et al., 2025), PaliGemma Beyer et al. (2024), and $\pi_{0.5}$ (Intelligence et al., 2025). Our models preserve strong performance across diverse multimodal understanding tasks despite only training on robot data. The best and second best results for each benchmark are shown in **bold** and underlined, respectively.

| Method | #Params | MMMU | MMStar | MME | OCRBench | MMB-en | MMB-cn | TextVQA | DocVQA | InfoVQA | AI2D | ChartQA | RealWorldQA |
|---|---|---|---|---|---|---|---|---|---|---|---|---|---|
| | | | | | Prismatic VLM Family | | | | | | | | |
| **Prismatic VLM** | 7b | 35.0 | 38.8 | **1456.6** | 32.0 | 66.2 | 55.7 | 42.5 | 17.5 | 19.7 | 54.6 | 16.7 | 30.8 |
| **OpenVLA** | 7b | 26.3 | 0 | 0 | 0 | 0 | 43.0 | 0 | 0 | 0 | 0 | 0 | 0 |
| **ECoT** | 7b | 26.6 | 0 | 0 | 0.01 | 3.7 | 4.1 | 0 | 0 | 0 | 0 | 0 | 25.6 |
| | | | | | Gemma-3 Family (with VLM2VLA) | | | | | | | | |
| **Gemma-3-4B-IT** | 4b | 39.3 | 37.1 | 1205.8 | 70.2 | 68.6 | 64.3 | 61.5 | 68.8 | 40.9 | 70.5 | 50.3 | 44.0 |
| **Gemma-3-12B-IT** | 12b | **46.0** | 46.3 | 1182.3 | **75.0** | 76.9 | 74.7 | 68.9 | 80.6 | 50.4 | 78.5 | 55.1 | 50.6 |
| **VLM2VLA-AT** | 12b | 45.9 | 45.2 | 1082.2 | 65.5 | 70.9 | 66.8 | 64.2 | 74.6 | 44.8 | 74.1 | 41.8 | 44.5 |
| **VLM2VLA (Ours)** | 12b | 42.7 | 48.0 | 1391.7 | 63.9 | 68.5 | 67.6 | 64.9 | 78.4 | 46.2 | 74.0 | 58.3 | 43.3 |
| | | | | | Molmo VLM Family | | | | | | | | |
| **Molmo** | 7b | 45.3 | **53.8** | 1275.3 | 69.5 | 65.9 | 61.3 | **81.7** | **92.2** | **72.6** | **93.2** | **84.1** | **70.7** |
| **MolmoAct** | 7b | 28.4 | 1.2 | 1224.5 | 52.7 | 55.1 | 46.3 | 57.5 | 58.7 | 41.9 | 2.0 | 55.9 | 8.6 |
| | | | | | PaliGemma VLM Family | | | | | | | | |
| **PaliGemma** | 3b | 26.1 | 49.0 | 1199.9 | 46.4 | 58.8 | 35.2 | 53.4 | 34.7 | 20.6 | 38.4 | 16.3 | 55.7 |
| **$\pi_{0.5}$** | 3b | 24.0 | 21.7 | 1061.9 | 6.8 | 6.8 | 0.3 | 10.0 | 4.6 | 7.7 | 27.0 | 5.1 | 2.7 |

**Q1–Multimodal Understanding:** Does VLM2VLA effectively preserve the base VLM's multimodal reasoning ability in standard VQA tasks after fine-tuning on robot data $\mathcal{D}_{\text{lan}}$?

**Q2–Robotic Manipulation:** Can VLM2VLA achieve a level of in-distribution manipulation performance that is competitive with state-of-the-art VLAs?

**Q3–Reasoning Generalization:** Does the preserved knowledge from Q1 directly enable VLM2VLA to generalize to novel, out-of-distribution robotics tasks that require reasoning beyond the fine-tuning data $\mathcal{D}_{\text{lan}}$?

**Baselines**: We contrast our policy against two state-of-the-art tokenization-based VLAs: OpenVLA (Kim et al., 2024), an autoregressive policy constructed from fine-tuning a pretrained 7B Prismatic VLM on the large-scale Open-X-Embodiment dataset (Karamcheti et al., 2024; Padalkar et al., 2024), and Embodied Chain-of-Thought (ECoT), a variant of OpenVLA trained to produce reasoning traces before action prediction (Zawalski et al., 2025). ECoT, similar to VLM2VLA, is only fine-tuned on the Bridgev2 dataset.

**Ablations**: We ablate our action representation by mapping the digits (0-9) in our robot action data to the decoded strings of Gemma-3's ten least likely tokens; therefore, during VLA fine-tuning, the model will learn to autoregressively predict actions via these reserved tokens (see Appendix B.3). We call this variation of our method, with action tokens, VLM2VLA-AT. To ensure an accurate comparison, both models are trained identically and trained on the same data, except for the action representation.

## 4.1 EVALUATION OF MULTIMODAL UNDERSTANDING

In this section, we quantify the extent to which VLM2VLA, and its variants, preserve their multimodal understanding by thorough evaluation on numerous VQA benchmarks. As demonstrated by Table 1, both OpenVLA and ECoT suffer from catastrophic forgetting relative to the original Prismatic VLM (Karamcheti et al., 2024; Zhou et al., 2025b). In contrast, VLM2VLA experiences only minor losses in performance across all VQA benchmarks, thereby conclusively answering Q1. While the VLM2VLA-AT ablation achieved comparable VQA scores, suggesting that our LoRA training scheme is primarily responsible for mitigating catastrophic forgetting, we argue this is a necessary but not sufficient condition for creating a generalist policy. As we show in Section 4.2, the choice of action representation is an important factor affecting generalization in downstream robotic tasks.

To compare VLM2VLA against state-of-the-art co-trained VLAs, we also evaluate two recently released open-source models: MolmoAct (Lee et al., 2025) and $\pi_{0.5}$ (Intelligence et al., 2025), along with their VLM backbones Deitke et al. (2024); Beyer et al. (2024). While MolmoAct is strong in VQA ability, VLM2VLA outperforms it across all reported benchmarks. We acknowledge that

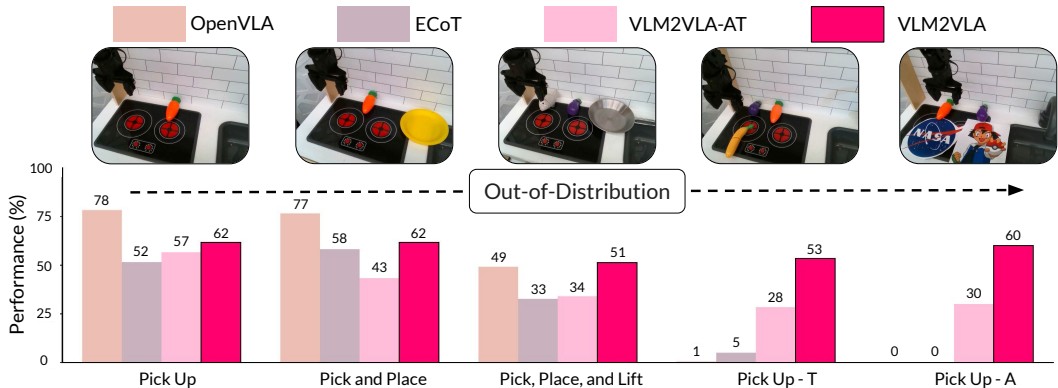

Figure 5: Comparative evaluation of VLA performance on in-distribution (ID) and out-of-distribution (OOD) robotic manipulation tasks. VLM2VLA maintains high success rates on OOD tasks, highlighting its superior generalization capabilities. Each bars corresponds to an average over thirty trials, except for the 'Pick Up - T' task, where each bar corresponds to an average over ninety trials.

model size is a confounding variable in the comparison, yet, the results in Table 1 demonstrate that our methodology is competitive with co-trained VLAs. On the other hand, the significantly lower performance of $\pi_{0.5}$ across most VQA benchmarks underscores that co-training is not a guaranteed solution to catastrophic forgetting.

## 4.2 EVALUATING ROBOTIC MANIPULATION

Having established that VLM2VLA and its variants retain their foundational pretraining knowledge, we now evaluate if this training procedure translates to robotic control. Specifically, we assess VLM2VLA's ability to perform robotic manipulation tasks of varying complexity in in-distribution (ID) and out-of-distribution (OOD) scenarios. All real-world policy evaluations receive a total of 30 trials and are performed on a 6-DoF WidowX 250S robotic arm in a toy kitchen environment, as prescribed in (Walke et al., 2024).

We compare VLM2VLA to the baselines and ablations on two ID tasks, one borderline ID task, and two OOD tasks. The in-distribution experiments probe each policy's action prediction capabilities over short and long horizons. The out-of-distribution experiments investigate if the policies can leverage the latent world knowledge of their VLM backbone to generalize beyond the VLA fine-tuning data. All skills tested are 'pick up' or 'pick and place' tasks, which are common in policy evaluation (Hancock et al., 2024; Snyder et al., 2025; Kim et al., 2024; Zawalski et al., 2025; Gao et al., 2025), and we describe each in detail below.

**Pick Up the Carrot (Pick Up)**: A baseline task testing the model's fundamental localization and grasping capabilities. This task is considered in-distribution because the task object (carrot) and instruction are within the VLA fine-tuning data.

**Put the Carrot On the Yellow Plate (Pick and Place)**: A longer horizon baseline task testing the model's ability to perform more complex actions. This task is considered in-distribution because the object and instruction are within the VLA fine-tuning data.

**Put the Eggplant In the Pan, Then Lift the Fish (Pick, Place, and Lift)**: A long-horizon, multi-step task requiring the robot to perform both pick-and-place and pick-up actions. This task is borderline in-distribution; the objects and instructions, when considered separately, are in-distribution, but not when combined.

**Pick Up the Carrot (Pick Up - T)**: A pick-up task with natural language instruction given in one of three languages: Spanish, Mandarin, or Hindi. This task is considered out-of-distribution because the instructions are not within the VLA fine-tuning dataset, thereby requiring the model to implicitly translate the task. Two distractor objects are present in each trial to prevent the policy from inferring the task from the observation alone.

**Pick Up the Item Above Ash Ketchum (Pick Up - A)**: A pick-up task requiring the model to first locate the pop-culture figure 'Ash Ketchum' and lift the correct object. This task is considered

out-of-distribution because VLAs are not trained to recognize semantic concepts like 'Ash Ketchum,' thereby requiring the model to leverage its latent world knowledge to succeed. Distractor logos ('NASA') and objects are present. The item located above 'Ash Ketchum' varies in-between trials, as does its location in the environment.

We report the full task success rates for all models in Fig. 5 (see Appendices D.1 to D.6 for additional scoring and experimental details). For the multilingual translation tasks (Pick Up -T), we report the average across all languages, which amounts to ninety trials per model.

### 4.2.1 RESULTS OF BASELINES

On the ID pick-up and pick-and-place tasks, OpenVLA performs the best. This high performance is likely explained by fine-tuning on the much larger Open-X-Embodiment dataset, which provides broader coverage of common grasping skills. Nevertheless, the comparable success rates observed for VLM2VLA answer Q2: the 'action as language' approach of VLM2VLA enables competitive action prediction in a specific robot embodiment.

As task complexity increases, the advantage of a purely reactive policy, like OpenVLA, starts to diminish. In the compositional task 'Put the Eggplant In the Pan, Then Lift the Fish', we found that OpenVLA often successfully completes the first subtask (put the eggplant in the pan), but fails to attempt the second. ECoT, on the other hand, normally generates motion plans to finish both subtasks, but fails to execute them as accurately as OpenVLA. We found that of the three policies, for this task, VLM2VLA not only generates correct hierarchical reasoning, but also executes them more frequently (see Fig. 5 and Fig. 6).

The OOD scenarios most clearly demonstrate the benefits of our knowledge-preserving pipeline. In the multilingual translation experiment (Pick Up - T), our method significantly outperforms both OpenVLA and ECoT. As shown in Fig. 6 and Fig. 7, VLM2VLA frequently translates the non-English language instruction and identifies the correct object; this is a direct application of the multilingual capabilities retained by our model as reported in Section 4.1. While ECoT does achieve a non-zero task success rate, the results presented in Fig. 6 suggest it may be relying on heuristics, such as simply picking up the closest or most salient object in the near vicinity, rather than a genuine understanding of the instruction.

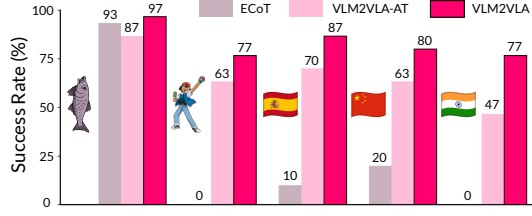

Figure 6: Analysis of task decomposition for OOD manipulation tasks, denoted by the corresponding icon. Points are awarded if the model's task plan correctly identifies the task object, and task destination, if present. See Appendix D.3 for additional details.

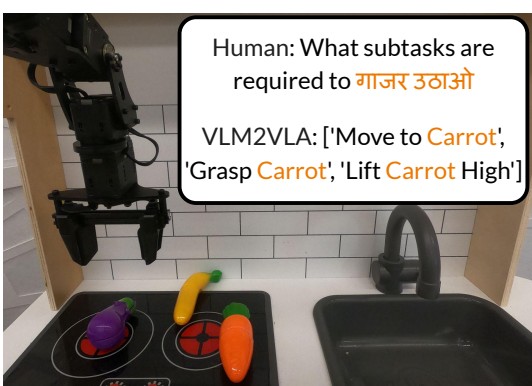

Figure 7: A qualitative demonstration of VLM2VLA's zero-shot multilingual capabilities. Given the language instruction in Hindi ('pick up the carrot'), our model identifies the correct object amidst distractors (eggplant and banana), demonstrating a genuine understanding of the task.

This trend is even more pronounced in the 'Pick Up the Item Above Ash Ketchum' task, requiring a combination of both open-world semantic knowledge (recognizing a pop-culture figure) and spatial reasoning. Here, VLM2VLA is the only model to achieve a meaningful success rate. This result provides a positive answer to Q3: the foundational world knowledge preserved by our method can be leveraged to solve novel problems zero-shot.

### 4.2.2 RESULTS OF ABLATIONS

Section 4.1 demonstrated that LoRA fine-tuning can mitigate catastrophic forgetting for both our method and the token-based ablation, but does the 'actions as language' representation actually facilitate learning a more effective policy? The results demonstrated in Fig. 5 and Fig. 6 show our approach consistently outperforms the token-

based ablation across all manipulation tasks. While VLM2VLA-AT performs competitively on simple ID tasks, its performance degrades as task complexity increases; this is seen most clearly in the OOD setting despite retaining strong VQA capabilities. For instance, Fig. 6 shows VLM2VLA-AT struggles with multilingual commands and scores only half as well as VLM2VLA in the 'Pick Up the Item Above Ash Ketchum' task (achieving a success rate of just 30% to our model's 60%). These findings are suggestive of a disconnect between the VLM's latent world knowledge and the action token assignment achieved after fine-tuning.

## 5 Conclusion and Discussion

In this work, we present VLM2VLA: a novel VLA training paradigm that addresses the challenge of catastrophic forgetting often experienced when adapting a VLM for robotic control. We demonstrate that by grounding the action prediction problem in natural language, LoRA can be used to train manipulation policies without significantly degrading the backbone VLM's foundational capabilities. Our data-centric approach of treating 'actions as language' obviates the need for major architectural modifications to the VLM or expensive co-training schemes. As our experiments show, the retained latent world knowledge of VLM2VLA enables zero-shot generalization to novel tasks that require open-world semantic reasoning. Ultimately, this work offers one path toward building generalist policies by closing the gap between large-scale foundation models and real-world robotic control.

### 5.1 Limitations and Future Work

Our approach has several limitations that present clear avenues for future work.

**Inference latency.** VLM2VLA generates actions autoregressively, which is inherently slow; in our experiments, the median run-time required for one cycle of action generation was 6.1 seconds, though subject to high variance (see Appendix D.6 for additional inference details). Future work could explore more advanced decoding techniques to accelerate our hierarchical prediction scheme without compromising the VLM's pretrained representations.

**Dexterous tasks.** Our work has focused on translational end-effector control, which is suitable for many simple pick-and-place tasks. While this design choice afforded strong action labeling from Gemini, it precludes dexterous manipulation tasks which require nuanced action prediction, e.g., rotation. Moreover, the granularity at which we predict motion plans is coarse, and a logical next step is to obtain fine-grained language annotations when constructing our VLA fine-tuning dataset. As vision-language models become more adept at spatial reasoning, we anticipate this re-labeling scheme will enable more precise robotic control.

**Cross-embodiment learning.** The scope of our work limited training to a specific robot embodiment, thereby preventing the same policy from readily adapting to other robots, especially those which utilize other means of low-level control, such as joint angles, which do not map easily to spatial affordances. Yet, we believe our 'actions as language' relabeling scheme is a promising solution. Using language as a common medium, it ought to be possible to describe all robot actions regardless of embodiment. Therefore, performing VLM2VLA's relabeling scheme could potentially be used to train a cross-embodiment policy.

**Improving or eliminating the verifier.** Our method currently relies on a separate verifier module to transition between subtasks, further slowing down inference speed. Future work could explore how to better train the base VLM as a verifier, thereby averting this step altogether.

**Larger-scale training.** We are working towards scaling our training pipeline with larger robotics datasets. We hypothesize that combining our language-based action representation with large-scale training may unlock zero-shot generalization, open-world semantic reasoning, and instruction following capabilities that go well beyond the state-of-the-art.

## Reproducibility Statement

All models reported in this work are open-source; our work, VLM2VLA, is based on the publicly available Gemma 3 model (Team et al., 2025b). We have attached an extensive Appendix providing additional details of our approach. Information regarding our data curation procedure is available in

Appendices B.1 to B.3. Our VQA analysis is listed in Appendix C. Our hardware experiments are described in Appendices D.1 to D.6. Full training details and the computational resources used are provided in Appendix E.

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

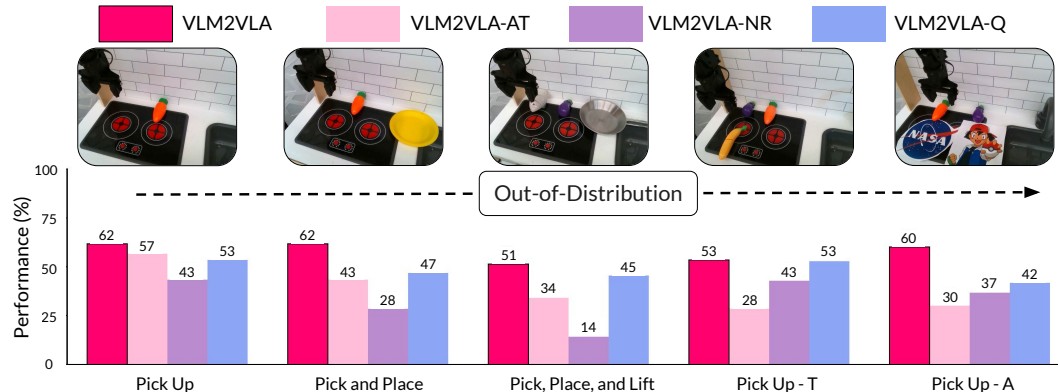

Figure 8: Comparative evaluation of VLM2VLA ablations on in-distribution (ID) and out-of-distribution (OOD) robotic manipulation tasks. VLM2VLA maintains high success rates on OOD tasks relative to ablated variants, highlighting the importance of reasoning and the robustness of our methodology across different model architectures. Each bar corresponds to an average over thirty trials, except for the 'Pick Up - T' task, where each bar corresponds to an average over ninety trials.

## A  ADDITIONAL EXPERIMENTS

To further investigate the core components of our methodology, we conduct two additional ablation studies examining: (1) how test-time reasoning impacts manipulation performance, and (2) if VLM2VLA generalizes to different VLM architectures. For the first ablation, we train VLM2VLA without the reasoning components (i.e., high-level subtask prediction and mid-level motion planning), denoting this variant as VLM2VLA-NR. For the second ablation, we apply our methodology to Qwen-3-8B-IT Yang et al. (2025a), denoting this model as VLM2VLA-Q. Both ablations are evaluated on the same suite of manipulation tasks described in Section 4.2, with results reported in Fig. 8. We additionally include the action-token ablation (VLM2VLA-AT) from Section 4.2 for completeness.

**Does Reasoning Impact Robotic Manipulation?** We investigate the contribution of test-time reasoning to VLM2VLA's performance by comparing against VLM2VLA-NR, an ablation trained without reasoning capabilities (i.e., subtask prediction and motion planning). As demonstrated in Fig. 8, the absence of reasoning substantially degrades performance on longer-horizon tasks. Notably, VLM2VLA-NR

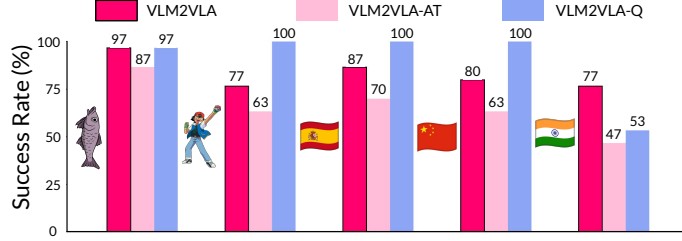

Figure 9: Analysis of task decomposition for OOD manipulation tasks, denoted by the corresponding icon. Points are awarded if the model's task plan correctly identifies the task object, and task destination, if present. See Appendix D.3 for additional details.

exhibits reduced success rates on pick-and-place tasks compared to the full VLM2VLA model. This performance gap is further amplified in OOD scenarios that require leveraging latent world knowledge, where VLM2VLA-NR achieves markedly lower success rates. These results underscore that the three-step reasoning procedure proposed in our method is critical for enabling effective manipulation, particularly in complex and novel task settings.

**Is VLM2VLA Model Agnostic?** To assess whether our 'actions as language' approach generalizes across different VLM architectures, we train VLM2VLA-Q using identical data and hyperparameters as our primary model, but with Qwen-3-8B-IT as the backbone instead of Gemma-3-12B-IT. As shown in Fig. 8, VLM2VLA-Q achieves competitive success rates across all tasks, demonstrating that our methodology is indeed model agnostic. While VLM2VLA-Q exhibits slightly lower overall task success rates compared to the full VLM2VLA model, this difference can be partially attributed to the reduced model capacity (8B vs. 12B parameters). Interestingly, Fig. 9 reveals that VLM2VLA-Q demonstrates comparable planning capabilities to VLM2VLA, successfully translating multilingual commands and correctly identifying task objects in scenarios requiring latent world knowledge. This

finding suggests that the gap in task success rates may stem from action execution precision rather than reasoning deficiencies. Nevertheless, the consistent performance of VLM2VLA-Q validates that our approach can be successfully applied to different VLM architectures.

## B  DATA CURATION

### B.1  PROMPTING

In this section, we showcase our Gemini prompting strategy to generate language-annotated robot trajectories at scale. Our example is specific to the WidowX 250S robot embodiment, focusing solely on translational degrees of freedom. All text within the prompt is invariant across robot trajectories, except for words/phrases designated in color, which are trajectory-dependent.

---

**Gemini Inputs and Prompt Template**

**Inputs:**

Gemini is provided with the robot trajectory data $\tau = \{(o_t, a_t)\}_{t=0}^{T}$ consisting of RGB observations $o_t$ and actions $a_t$ controlling the relative position of the robot's end-effector. Additionally, Gemini is provided with the main-task instruction $L$. We combine all actions and instruction into a text file called trajectory_log_content.

**Gemini Prompt**

Attached are images of a robot performing a task, and the real actions in a text file called trajectory_log_content. You are programming a 4-DoF robot arm (x,y,z,gripper). Positive x points forward (negative x points backward) with respect robot's end-effector. Positive y points left (negative y points right) with respect to the robot's end effector. Positive z points up (negative z points down) with respect to the robot's end-effector. The gripper takes a binary value of 1 for open and 0 for closed.

Your task is to analyze the language instruction found in the trajectory log. Crucially, first think step-by-step and explain your reasoning for each action phase necessary to complete the instruction given all the images and their ordering during the trajectory. Be short and concise. Output your reasoning as a structured JSON file.

Then, after you've explained your reasoning, give the actions with respect to (dx, dy, dz, gripper) that the robot must execute to achieve this goal given the orientation described previously, associating each action or group of actions with the natural language decomposition you decided on. Store these actions in JSON format as a list of dictionaries, where each dictionary has 'step_description' (string) and 'actions' (list of lists representing [dx, dy, dz, gripper]). Make sure your actions match what is provided in the attached .txt file. For example:

```
{{
"step_description": "Grasp the Pepper",
"actions": [
      [0.010, 0.026, 0.002, 1.0],
      [0.017, 0.033, 0.008, 1.0],
      [0.003, 0.007, -0.018, 1.0],
      [-0.006, 0.0, -0.015, 1.0],
      [0.003, 0.002, 0.001, 0.0]
    ]
}}
```
Here is the trajectory log content: trajectory_log_content

---

In summary, Gemini 2.5 Team (2025) is given the entire trajectory $\tau = \{(o_t, a_t)\}_{t=0}^{T} \in \mathcal{D}_{\text{rob}}$ and main-task instruction $L$ as context, as well as the robot's coordinate system and sign convention in the base frame. For example, the Bridgev2 dataset Walke et al. (2024) utilizes the 6-DoF WidowX 250S robotic arm, where movement in the positive 'x' direction corresponds to 'forward', negative movement in the 'y' directions corresponds to 'right,' and so forth. Since all action sequences are given as relative movements of the robot's end-effector, the aforementioned coordinate system provides strong spatial grounding for Gemini to construct high-quality language annotations and determine the most appropriate timesteps to carve the trajectory into sub-trajectories. To address potential concerns about auto-labeling errors, we manually inspected a random subset of Gemini's generated data for accuracy in subtask decomposition, motion-planning, and action prediction.

Initially, we solely used Gemini 2.5 Pro, but later switched to Gemini 2.5 Flash to save on costs. In total, generating the entire dataset costed roughly $900. Finally, we provide an example of single language-annotated trajectory from the Bridgev2 dataset. We additionally include the first and last observations of the robot trajectory for reference.

Figure 10: Key observations throughout the original, robot trajectory, which are provided to Gemini as context.

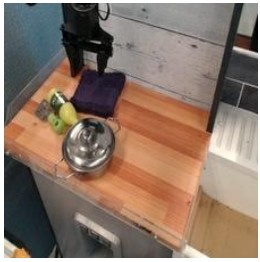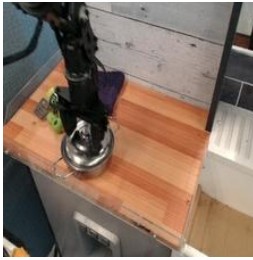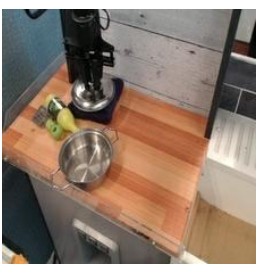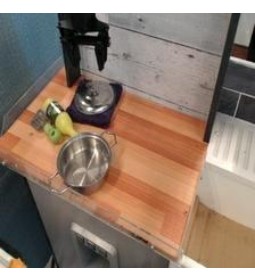

(a) Observation: 0 (Initial)     (b) Observation: 10     (c) Observation: 20     (d) Observation: 28 (Final)

**Gemini Language Annotation Example**

**Output:**

Below is the structured JSON output generated by Gemini for the task of $L =$ 'open the silver pot'. Gemini first decomposes the trajectory into sequence of steps, then performs a chain-of-thought reasoning to generate a motion plan. Finally, it copies the corresponding actions to each step.

```json
[
  {
    "STEP_DESCRIPTION": "Move to Lid Handle",
    "REASONING": "The robot must first navigate its
        end-effector to be directly above the lid handle. This
        requires moving downwards (-dz) and adjusting
        the  horizontal position (likely backward -dx and right
        -dy based on the initial position and target). The
        gripper must remain open (1) for grasping.",
    "MAIN_MOVEMENTS": "Primarily backward (-dx), right (-dy),
        and downward (-dz) movements. Gripper stays open (1).",
    "ACTIONS": [
      [ 0.0,    0.0,    0.0,    1.0 ],
      [ -0.002,  -0.002,  -0.013,  1.0 ],
      [ -0.023,  -0.007,  -0.005,  1.0 ],
      [ -0.018,  -0.006,  0.022,  1.0 ],
      [ -0.018,  -0.005,  -0.003,  1.0 ],
      [ -0.018,  -0.006,  -0.005,  1.0 ],
      [ -0.015,  -0.014,  -0.014,  1.0 ],
```

```
        [ -0.004,  -0.010,  -0.007,  1.0 ],
        [ 0.0,  -0.005,  -0.003,  1.0 ]
      ]
    },
    {
      "STEP_DESCRIPTION": "Grasp Lid Handle",
      "REASONING": "Once positioned correctly, the robot needs to
          grasp the handle. The robot closes the gripper (0).
          There might be a final small downward adjustment (-dz)
          just before or during the grasp.",
      "MAIN_MOVEMENTS": "Close gripper (0), minor downward
          adjustment (-dz). Gripper changes to 0.",
      "ACTIONS": [
        [ 0.002,  -0.007,  -0.010,  0.0 ]
      ]
    },
    {
      "STEP_DESCRIPTION": "Lift Lid",
      "REASONING": "After securing the handle, the robot must
          lift the lid vertically (+dz) to clear the base of
          the pot. It also needs to move it slightly away
          from the pot base, involving some horizontal movement
          (likely forward +dx and left +dy). The gripper remains
          closed (0) to hold the lid.",
      "MAIN_MOVEMENTS": "Primarily upward (+dz), forward (+dx), a
          nd left (+dy). Gripper remains closed (0).",
      "ACTIONS": [
        [ 0.007,  0.0,  -0.001,  0.0 ],
        [ 0.002,  0.0,  0.0,  0.0 ],
        [ -0.002,  -0.002,  0.0,  0.0 ],
        [ -0.004,  0.0,  0.002,  0.0 ],
        [ 0.0,  0.004,  0.004,  0.0 ],
        [ 0.015,  0.014,  0.017,  0.0 ],
        [ 0.023,  0.007,  0.020,  0.0 ]
      ]
    },
    {
      "STEP_DESCRIPTION": "Move Lid to Placement Spot & Lower",
      "REASONING": "The goal is to open the pot, implying the lid
          needs to be placed somewhere else. This involves
          horizontal movement (forward +dx, left +dy) towards the
          placement spot and then lowering the lid (-dz). The
          gripper stays closed (0).",
      "MAIN_MOVEMENTS": "Primarily forward (+dx), left (+dy), and
          downward (-dz). Gripper remains closed (0).",
      "ACTIONS": [
        [ 0.027,  0.008,  -0.003,  0.0 ],
        [ 0.021,  0.017,  -0.007,  0.0 ],
        [ 0.018,  0.007,  -0.011,  0.0 ],
        [ 0.015,  0.007,  -0.010,  0.0 ],
        [ 0.010,  0.004,  -0.014,  0.0 ]
      ]
    },
    {
      "STEP_DESCRIPTION": "Release Lid",
      "REASONING": "Once the lid is resting on the surface, the
          robot must release it by opening the gripper (1).",
      "MAIN_MOVEMENTS": "Open gripper (1).",
```

```
    "ACTIONS": [
      [ 0.003,  0.001,  -0.002,  1.0 ]
    ]
  },
  {
    "STEP_DESCRIPTION": "Retract",
    "REASONING": "Finally, the robot should move its arm away
        from the lid and the pot to a safe, neutral position,
        typically upwards (+dz) and potentially backward
        (-dx). The gripper remains open (1).",
    "MAIN_MOVEMENTS": "Primarily upward (+dz) and backward
        (-dx). Gripper remains open (1).",
    "ACTIONS": [
      [ -0.002,  -0.002,  0.0,  1.0 ],
      [ -0.005,  -0.005,  0.006,  1.0 ],
      [ 0.0,  -0.003,  0.022,  1.0 ],
      [ -0.010,  0.002,  0.022,  1.0 ],
      [ -0.010,  -0.001,  0.017,  1.0 ],
      [ 0.0,  0.0,  0.001,  1.0 ]
    ]
  }
]
```

## B.2 DATA PREPARATION

After Gemini splits each trajectory into sub-trajectories, we found it helpful in policy training to chunk the actions along a single dimension to have larger magnitudes; in particular, we found that without this step, VLM2VLA often provided negligible action predictions, even though its predicted motion-plans appears sensible.

In this work, we used a threshold along a single dimension, without sign change, of 2.5 centimeters and an absolute threshold of 5 centimeters. We include an example of the data for training Gemma-3 before and after this post-processing step below:

### Original Sub-Trajectory Action Data

**Inputs:**

The model is provided with the subtask 'Grasp the Yellow Pepper' and image file: obs_0.jpg.

**Human Prompt**

You are programming a 4-DoF robot arm. Based on the image (obs_0.jpg) and subtask 'Grasp the Yellow Pepper', output the sequence of low-level [x, y, z, gripper] actions needed to accomplish the task.

**Model Output**

```
[[ 0.000,   0.000,   0.000, 1.0],
 [-0.002,   0.000,  -0.007, 1.0],
 [ 0.000,  -0.004,  -0.016, 1.0],
 [ 0.002,  -0.002,  -0.014, 1.0],
 [ 0.003,   0.000,  -0.008, 1.0],
 [ 0.002,   0.000,  -0.011, 1.0],
 [ 0.000,   0.000,  -0.005, 1.0],
 [ 0.000,   0.000,  -0.007, 1.0],
 [ 0.000,   0.000,  -0.006, 1.0],
 [ 0.001,  -0.003,  -0.003, 0.0]]
```

**Post-Processed Sub-Trajectory Action Data**

**Inputs:**

The model is provided with the subtask 'Grasp the Yellow Pepper' and image file: obs_0.jpg

**Human Prompt**

You are programming a 4-DoF robot arm. Based on the image (obs_0.jpg) and subtask 'Grasp the Yellow Pepper', output the sequence of low-level [x, y, z, gripper] actions needed to accomplish the task.

**Model Output**

```
[[-0.002,  -0.004,  -0.023, 1.0],
  [0.007,  -0.002,  -0.045, 1.0],
  [0.0, 0.0,  -0.006, 1.0],
  [0.001,  -0.003,  -0.003, 0.0]]
```

We further augment our dataset with two additional sets of data. First we construct positive and negative examples of subtask completion, each consisting of a tuple $(\bar{o}_i, \bar{o}_j, l_i)$ with $j$ distinct from $i$: the robot successfully completed subtask $i$ if and only if $j = i + 1$. In principle, this data could train the VLM as the verifier, but we found our model unable to reliably perform this step. We leave addressing this shortcoming as future work.

Secondly, we train the model to predict which direction (e.g., left, right, or none) the robot should move to complete a given subtask, using the same movement threshold of 2.5 centimeters. While we do not query our VLA in this manner at run-time, we found this auxiliary training signal beneficial for downstream performance.

### B.3 ACTIONS AS TOKENS

For the action representation ablation, we cast the digits 0-9 to the tenth least likely tokens under the Gemma-3-12B-IT model. We describe this mapping in the table provided in Table 2. In addition, we provide an example of decoded model output under this mapping, identical to the one provided above.

Table 2: Mapping of digits to the least likely tokens under the Gemma-3 model.

| Digit | Token | Token ID |
|---|---|---|
| 9 | <unused6142> | 262044 |
| 8 | <unused6141> | 262043 |
| 7 | <unused6140> | 262042 |
| 6 | <unused6139> | 262041 |
| 5 | <unused6138> | 262040 |
| 4 | <unused6137> | 262039 |
| 3 | <unused6136> | 262038 |
| 2 | <unused6135> | 262037 |
| 1 | <unused6134> | 262036 |
| 0 | <unused6133> | 262035 |

**Post-Processed Sub-Trajectory Action Data Under Action Tokenization**

**Model Output**

```
[[-<unused6133>.<unused6133><unused6133><unused6135>, -<u
nused6133>.<unused6133><unused6133><unused6137>, -<unused
6133>.<unused6133><unused6135><unused6136>, <unused6134>.
<unused6133>],
 [<unused6133>.<unused6133><unused6133><unused6140>, -<un
used6133>.<unused6133><unused6133><unused6135>, -<unused6
133>.<unused6133><unused6137><unused6138>, <unused6134>.<
unused6133>],
 [<unused6133>.<unused6133>, <unused6133>.<unused6133>, -
<unused6133>.<unused6133><unused6133><unused6139>, <unuse
d6134>.<unused6133>],
 [<unused6133>.<unused6133><unused6133><unused6134>, -<un
used6133>.<unused6133><unused6133><unused6136>, -<unused6
133>.<unused6133><unused6133><unused6136>, <unused6133>.<
unused6133>]]
```

## C  VQA Experiment Details

We build our evaluation pipeline using the open-source framework lmms-eval[1], which provides standardized implementations of multimodal benchmark datasets, evaluation metrics, and inference wrappers for a wide range of models. We report results for our VLM2VLA model (finetuned with Gemma-3-12B backbone), two strong baselines (OpenVLA and ECoT, both Prismatic-7B backbone), and the original instruction-tuned Gemma-3-4B and Gemma-3-12B models.

MolmoAct and $\pi_{0.5}$ are designed to handle inputs from multiple cameras (e.g., exterior and wrist views). However, since standard VQA benchmarks provide only a single image per sample, we use the multi-view vision-language model to process single-image VQA benchmarks by utilizing only the primary image input from each sample and masking the other image inputs. We use the $\pi_{0.5}$-base checkpoint for all $\pi_{0.5}$ experiments.

## D  Robot Manipulation Tasks and Detailed Results

### D.1  Experiments and Manipulation Evaluation Metrics

As described in Section 4.2, we evaluate all generalist robot policies on two in-distribution tasks, one borderline in-distribution task, and two out-of-distribution tasks. Each policy received a total of 30

---

[1]https://github.com/EvolvingLMMs-Lab/lmms-eval

roll-outs for each task. In this section, we provide additional details on the scoring rubric used to evaluate each method, giving special care to fairly evaluate planning and non-planning policies.

**Pick Up the Carrot**: The robot's goal is to navigate to and lift up the carrot. There are no distractor objects present. This experiment is considered within distribution because it is part of the Bridgev2 dataset.

Between trials, the carrot's initial position varied between three possibilities: (i) directly below the robot's gripper, (ii) directly in front of the robot, (iii), in front and to the left or right of the robot. The robot's initial position was constant between trials.

*Scoring*: Partial credit (1 out of 2) is assigned if the robot makes contact with the carrot.

**Put the Carrot On the Yellow Plate**: The robot's goal is to grasp the carrot, move to the plate, and successfully release it on the plate. There are no distractor objects present. This experiment is considered within distribution because it is part of the Bridgev2 dataset.

Between trials, the carrot's initial position varied between two possibilities: (i) directly in front of the robot and (ii) in front and to the left or right of the robot. The robot's initial position was constant between trials.

*Scoring*: Partial credit (1 out of 2) is assigned if the robot makes contact with the carrot.

**Put the Eggplant In the Pan, Then Lift the Fish**: The robot's goal is to grasp the eggplant, move and release it in the pan, then navigate to and successfully lift the fish. This task is considered borderline in-distribution because the individual subtasks are within distribution, but the combination is not.

The eggplant is located to the right of the fish between all trials, which is located in front of and to the right of the robot's initial condition; all starting positions for the eggplant, pan, fish, and robot are invariant across trials.

*Scoring*: For non-planning policies (OpenVLA), partial credit (1 out of 5) is awarded if the robot makes contact with the eggplant, (2 out of 5) if the robot places the eggplant in the pan, (3 out of 5) if the robot moved toward the fish, (4 out of 5) if the robot contacts the fish, and (5 out of 5) if the robot lifts the fish.

For planning policies (ECoT, VLM2VLA), partial credit (1 out of 5) is awarded if the policy generates a subtask decomposition or motion-plan with the correct primitives: grasping the eggplant, moving to the pan, lifting the fish. The remaining partial credit is awarded the same as for non-plannign policies: (2 out of 5) if the robot makes contact with the eggplant, (3 out of 5) if the robot places the eggplant in the pan, (4 out of 5) if the robot contacts the fish, and (5 out of 5) if the robot lifts the fish.

**Pick Up the Carrot (Multilingual Translation)**: The robot's goal is to navigate to and lift up the carrot. Instructions are given in one of three languages: Spanish, Mandarin, and Hindi. This task is considered out of distribution because the language commands are not within the Bridgev2 dataset.

Between trials, the carrot's initial position varied between three possibilities: (i) directly below the robot's gripper, (ii) directly in front of the robot, (iii), in front and to the left or right of the robot. The robot's initial position was constant between trials. Two distractor objects (banana and eggplant) are present in each trial in an attempt to prevent the policy from inferring the task from the observation alone.

*Scoring*: For non-planning policies (OpenVLA), partial credit (1 out of 2) is awarded if the robot makes contact with the carrot. For planning policies (ECoT, VLM2VLA), partial credit (1 out of 2) is awarded if the policy's planning included the correct item 'carrot' *and* the robot makes contact with the carrot.

**Pick Up the Item Above Ash Ketchum**: The robot's goal is to navigate to and lift up the object situated above the pop-culture figure 'Ash Ketchum.' This task is considered out of distribution because the picture of 'Ash Ketchum' is not present in the Bridgev2 dataset.

Between trials, the picture of 'Ash Ketchum' varied between being either to the left or right of the robot's starting position, which remained invariant across trials. The target object is placed a few centimeters above the picture to preclude partial observability.

*Scoring*: For non-planning policies (OpenVLA), partial credit (1 out of 2) is awarded if the robot makes contact with the correct object. For planning policies (ECoT, VLM2VLA), partial credit (1 out of 2) is awarded if the policy's planning included the correct item *and* the robot makes contact with the item.

## D.2  HARDWARE DETAILS

All real-world policy evaluations were performed on a 6-DoF WidowX 250S robotic arm in a toy kitchen environment, as prescribed in (Walke et al., 2024). Images were captured with a Realsense D435 camera mounted on the right side of the robot to provide a third-person point of view. The initial conditions of task and distractor objects between trials were varied to ensure an accurate appraisal of policy performance. Like the environment, all objects are drawn from the Bridgev2 dataset to minimize out-of-distribution failures. To prevent damage to our setup, a safety filter was applied to all policies evaluated in this work. The filter prevented further execution of actions which would drive the robot into the environment, e.g., zero out downward or forward commands if the end-effector was contacting the counter or wall.

## D.3  TASK DECOMPOSITION EVALUATION METRICS

In this section, we describe the scoring procedure used to generate Fig. 6. The objective was to quantify each policy's ability to interpret and break down a variety of user instructions. Performance was measured by analyzing the textual output of each model's chain-of-thought reasoning, or subtask decomposition, whenever available.

For each task considered from Appendix D.1, a model's reasoning was considered successful if the model's textual output contained a specific set of required keywords. The evaluation was case-insensitive and manually checked to ensure all policies received a fair appraisal. The keywords and synonyms are defined as follows:

**eggplant**: purple, aubergine

**carrot**: orange, gajar

**pan**: (no synonyms defined)

**fish**: (no synonyms defined)

We now specify the scoring rubric for each experiment which required analysis:

**Put the Eggplant In the Pan, Then Lift the Fish**: success required the simultaneous presence of synonyms for 'eggplant,' 'pan,' and 'fish.'

**Pick Up the Carrot (Multilingual Translation)**: success required the presence of a carrot synonym in the model's output. Interestingly, some subtask decompositions from VLM2VLA occasionally contained the word 'gajar,' the Hindi synonym for carrot, demonstrating direct translation abilities of our method. This phenomena was not observed for Spanish nor Mandarin inputs.

**Pick Up the Item Above Ash Ketchum**: success required identifying the correct object for that specific trial. The required keyword was either a synonym of carrot *or* eggplant, depending on the ground truth object for that trial.

## D.4  TEST-TIME PROMPTING

In this section, we describe the prompting strategies used in VLM2VLA to elicit grounded subtask predictions, spatial reasoning for motion-planning, and translational action prediction.

First, given the initial image, we prompt VLM2VLA to break the main language instruction into a series of subtasks.

> **Subtask Prediction Prompt**
>
> Describe the sequence of remaining high-level steps required to complete the overall task 'main task', starting from the current state. Give your output as a list. Be specific and do not skip steps. Here is an example for 'put the pot in the sink': ['Move Down to Pot', 'Grasp Pot', 'Lift Pot High', 'Move Pot Left to the Sink', 'Lower Pot to Sink', 'Release']. Start with 'Move to'

For a given subtask, we prompt VLM2VLA with the current observation and subtask.

> **Motion-Planning Prompt**
>
> Given the image, reason about what high-level actions the robot arm should take to complete the task "subtask". +dx is forward, -dx is backward, +dy is left, -dy is right, +dz is up, -dz is down, 1 is gripper open, 0 is gripper closed. Provide concise, accurate spatial reasoning.

Once a motion-plan is generated, VLM2VLA is prompted with that reasoning, current observation, and subtask.

> **Action Generation Prompt**
>
> You are programming a 4-DoF robot arm (x,y,z,gripper). +dx is forward, -dx is backward, +dy is left, -dy is right, +dz is up, -dz is down, 1 is gripper open, 0 is gripper closed. Based on the sub-task 'subtask' and the reasoning 'motion-plan reasoning', output the sequence of low-level [dx, dy, dz, gripper] actions needed as a python list of lists. Be concise in direction of movement. Example format: [[dx1,dy1,dz1, grip1], [dx2,dy2,dz2, grip2]]. PROVIDE ONLY THE PYTHON LIST.

All outputs from VLM2VLA are generated using nucleus (top-p) sampling with p=0.95. We use distinct temperature settings for each generation stage: 0.1 for motion planning and 0.5 for action prediction. For the initial sub-task decomposition stage, the temperature is set to 0.5 for in-distribution tasks and 1.0 for all other scenarios. The aforementioned hyperparameters were empirically found to work well.

We use the default setting of greedy decoding for both the OpenVLA and ECoT baselines. The task prompts varied slightly between policies to maximize performance of each model on a given task. While all models received near identical commands, we followed the prompt formats suggested by (Kim et al., 2024; Zawalski et al., 2025) for the baselines, and used our own for VLM2VLA and VLM2VLA-AT (see Appendix D.4).

## D.5 Verifier Prompting

To improve robustness of the policy, we operate in closed-loop fashion with a verifier $V : \mathcal{O} \times \mathcal{O} \times \mathcal{L} \times \mathcal{L} \to \mathcal{L}$, where $\mathcal{O}$ is the RGB image space and $\mathcal{L}$ is the language space; at the end of each action-generation cycle, we query the verifier with the observations before ($\bar{o}_i$) and after ($\bar{o}_{i+1}$) action execution ($\bar{a}_i$), as well as the current ($l_i$) and next ($l_{i+1}$) subtask. The verifier $V(\bar{o}_i, \bar{o}_{i+1}, l_i, l_{i+1})$ will reason if the current subtask was completed successfully, given the observations and the next subtask. The output from the verifier dictates the subtask used in the next action-generation cycle, which continues until all $N$ subtasks are completed. In this work, we utilize Gemini 2.5 Pro (Team, 2025) as the verifier, albeit one may train the model itself to do this step.

In this section, we describe the prompting strategies used in this work to query the external verifier if the subtask was successfully completed.

**Verifier Prompting Strategy**

**Inputs:**

After having executed action chunk $\bar{a}_i$, Gemini is provided with two observations, $(\bar{o}_i, \bar{o}_{i+1})$ and the current and next subtask $(l_i, l_{i+t})$.

**Verifier Instructions**

You are a precision-oriented robot inspector. Your primary job is to evaluate the two images $(\bar{o}_i, \bar{o}_{i+1})$ if the 'Current Subtask' $(l_i)$ was executed with enough precision to guarantee the success of the 'Next Subtask' $(l_{i+1})$.

CRITICAL RULE: Judge the Precondition.

A 'move to' action is ONLY successful if the final position is perfectly aligned for the subsequent action, i.e., executing a grasp as the next step will result in a successful grasp. A 'close enough' position is a FAILURE if it jeopardizes the next step.

For a 'grasp' subtask to succeed, the gripper MUST be centered directly above the object's grasp point in the preceding 'move' step. Any significant offset or misalignment in the 'after' image is a failure.

Provide your analysis in the specified JSON format, following the turn-based examples.

**Expected Model Output (JSON)**

```
{
    "success": true,
    "confidence": "High",
    "reasoning": "This is a successful grasp because the
    second image clearly shows the robot has successfully
    grasped the eggplant."
}
```

## D.6 INFERENCE LATENCY

We conducted a series of rollouts and measured the wall-clock time for a single inference cycle, which we defined as the generation of both the mid-level reasoning trace and low-level action prediction. All experiments were conducted on an NVIDIA A100 GPU. We collected data across 30 evaluation runs and report the summary statistics in Table 3.

Table 3: Inference Latency Statistics (N=30 trials)

| Statistic | Value |
|---|---|
| Median | 6.1 [s] |
| Mean (Average) | 10.5 [s] |
| Standard Deviation | 14.3 [s] |
| Interquartile Range (IQR) | 5.0 - 6.7 [s] |
| Minimum | 3.8 [s] |
| Maximum | 48.8 [s] |

The median of 6.1 seconds and small interquartile range demonstrate VLM2VLA is capable of fast inference. However, the high standard deviation of 14.3 seconds warrants discussion. Our analysis of the output logs revealed that 1) approximately 10% of cases triggered our retry mechanism due to

bad output formatting from our model, causing additional run-time and 2) a small subset of trials exhibited unusually long run-times ($> 45$ seconds), suggesting a computational bottleneck.

These results highlight the importance of developing fast decoding schemes for low-latency performance in real-world scenarios, which we reserve for future work.

## E  TRAINING DETAILS

We fine-tune the Gemma-3-12B-IT (Team et al., 2025b) model using the TRL library (von Werra et al., 2025). Training is managed with Hugging Face's Accelerate framework, utilizing a DeepSpeed ZeRO Stage 2 configuration for efficient memory usage across multiple GPUs. We employ parameter-efficient fine-tuning (PEFT) using low-rank adaptation (LoRA) to update the model weights.

To obtain the final checkpoint, we train the model on a single node with 4 NVIDIA A100 GPUs. The final model was fine-tuned for one epoch, which we found sufficient for this work. In total, our model required approximately 300 GPU hours. The key hyper-parameters used for this training run are described in Table 4; notably, we found that using a small effective batch size worked well.

Table 4: VLM2VLA Training Hyper-parameters

| Hyperparameter | Value |
| --- | --- |
| **Training Strategy** | |
| Base Model | Gemma-3-12B-IT |
| Frameworks | TRL, Accelerate, DeepSpeed (ZeRO Stage 2) |
| Fine-Tuning Method | PEFT (LoRA) |
| Precision | bfloat16 (BF16) |
| **LoRA (PEFT) Configuration** | |
| Rank ($r$) | 16 |
| Alpha ($\alpha$) | 32 |
| Target Modules | `q_proj, k_proj, v_proj, o_proj, up_proj, down_proj, gate_proj` |
| **Optimization** | |
| Optimizer | AdamW |
| Learning Rate | 5e-5 |
| LR Scheduler | Linear Decay |
| Adam Beta1 | 0.9 |
| Adam Beta2 | 0.999 |
| Adam Epsilon | 1e-8 |
| **Dataloader Configuration** | |
| Global Batch Size | 1 |
| Per-Device Batch Size | 1 |
| Gradient Accumulation Steps | 2 |
| Effective Global Batch Size | 8 |
| Max Sequence Length | 1024 |

## F  USAGE OF LARGE LANGUAGE MODELS (LLMS)

During the preparation of this work, the authors utilized Large Language Models (LLMs) to support various aspects of the research and writing process. The specific applications of LLMs in this paper are outlined below:

**Code Generation and Debugging**: LLMs were employed to generate and debug code in the preliminary stages of this work. This included creating Python scripts to generate the plots presented

herein. Moreover, LLMs assisted in debugging error logs when training VLM2VLA and proposing the corresponding fixes.

**Manuscript Preparation**: LLMs were utilized to format tables in LaTeX code for use in this manuscript. Additionally, LLMs aided in creating the template used in the Appendix to designate the 'Human Prompt' and the 'Model Output.'

**Writing Assistance**: LLMs were queried to enhance the quality and readability of the manuscript. This involved checking grammar, and suggesting edits to improve sentence structure, syntax, and clarity, thereby helping to polish the final version.

