# OpenReview forum: "Actions as Language: Fine-Tuning VLMs into VLAs Without Catastrophic Forgetting"
_ICLR.cc/2026/Conference — ICLR 2026 Poster_

### Official Review · Reviewer_HTUd · 2025-10-29

**Soundness:** 4
**Presentation:** 3
**Contribution:** 4
**Rating:** 8
**Confidence:** 3

**Summary:**

The paper introduces *VLM2VLA* to overcome the trade-off where training a powerful VLM to control a robot leads to a loss of its general understanding. By first translating robot actions into natural language, VLM2VLA bridges the gap between the VLM's pretraining and the robot data. This technique allows the model to be efficiently fine-tuned with LoRA, keeping its original knowledge intact. This successful preservation is experimentally validated by the robot's ability to demonstrate advanced semantic reasoning and zero-shot generalization on novel, complex tasks.

**Strengths:**

- The paper is well-written and structured.
- The problem is clearly stated and is highly interesting and relevant to the robot learning community.
- The solution is quite novel. The idea of using natural language to describe actions is interesting and exceptionally well-presented and motivated in the paper, particularly in Figure 3.
- The experimental setup is strong; however, the results are most fascinating in the Out-of-Distribution (OOD) tasks, especially those where instructions were given in different languages. This particular study provides strong evidence that the *VLM2VLA* method effectively preserves the VLM's capabilities.
- The limitations and future work are clearly stated.

**Weaknesses:**

While this work presents many positive aspects, I offer a few suggestions that could further strengthen its quality and experimental validation.
- The baselines $\pi_{0.5}$ [1] and MolmoAct  [2] should also be evaluated on the real-world tasks for completeness, consistent with the other baselines in Section 4.2. Including these results would significantly strengthen the experimental validation and make the overall conclusions more robust.
- I believe the performance on the VQA tasks for the corresponding reference VLM models for $\pi_{0.5}$ [1] and MolmoAct [2] should be added as done with OpenVLA by including Prismatic VLM and VLM2VLA by including Gemma-3.
- While the chosen tasks are compelling and relevant, the overall number of experimental tasks discussed in the paper remains limited.

[1] Intelligence, Physical, et al. "$\pi_ {0.5} $: a Vision-Language-Action Model with Open-World Generalization." arXiv preprint arXiv:2504.16054 (2025).

[2] Lee, Jason, et al. "Molmoact: Action reasoning models that can reason in space." arXiv preprint arXiv:2508.07917 (2025).

**Questions:**

I have a couple of questions regarding the experimental methodology:
- Have you ablated different VLM models for the purpose of task curation?
- Was the robot learning fine-tuning based exclusively on the Bridgev2 dataset, or did it also include real demonstrations collected from the exact kitchen setup depicted in Figure 5?
- I am also interested in whether the In-Distribution (ID) tasks are considered ID solely based on the instruction and objects, or if they also rely on the identical physical kitchen setup and object locations.

---

> ### Author Response · Authors · 2025-11-21
>
> We thank the reviewer for their thorough review and positive feedback. In response to the reviewer’s noted weaknesses and questions, we address each below:
>
> > The baselines $\pi_{0.5}$ [1] and MolmoAct [2] should also be evaluated on the real-world tasks for completeness, consistent with the other baselines in Section 4.2. Including these results would significantly strengthen the experimental validation and make the overall conclusions more robust.
>
> We do not compare our method to these models in the hardware experiments because they both co-train on multimodal web datasets, and our method does not, thereby not offering a fair comparison. Moreover, running either model on our hardware setup requires a small amount of fine-tuning data, despite training on Bridgev2, thereby providing privileged information to the baselines.
>
> > I believe the performance on the VQA tasks for the corresponding reference VLM models for $\pi_{0.5}$ [1] and MolmoAct [2] should be added as done with OpenVLA by including Prismatic VLM and VLM2VLA by including Gemma-3.
>
> Thank you for your comment and we agree that providing the VQA evals for the VLM backbones of $\pi_{0.5}$ and MolmoAct could strength the paper. We have included this analysis in Table 1 in the updated PDF (see Table 1).
>
> > While the chosen tasks are compelling and relevant, the overall number of experimental tasks discussed in the paper remains limited.
>
> While we agree that the number of *different tasks and environments* considered in this paper was limited, we would like to highlight that we conducted over 800 real-world policy evaluations in producing this work, which is a time intensive endeavor. Each task required 30 evaluations to draw meaningful conclusions, as too few roll-outs could lead to statistically insignificant results.
>
> > Have you ablated different VLM models for the purpose of task curation?
>
> We did not ablate different VLMs in our data generation pipeline, primarily because we found that Gemini 2.5 was strong enough to do the autolabeling for us.
>
> > Was the robot learning fine-tuning based exclusively on the Bridgev2 dataset, or did it also include real demonstrations collected from the exact kitchen setup depicted in Figure 5?
>
> Yes, the VLA pretraining dataset consisted solely of the Bridgev2 dataset (see Section 3.2). No real-world demonstrations were used in fine-tuning our model for real-world deployment; this is partly why we didn’t compare with $\pi_{0.5}$ and MolmoAct, which would require a fine-tuning dataset for non-zero success rates, despite both models pretraining on Bridgev2.
>
> > I am also interested in whether the In-Distribution (ID) tasks are considered ID solely based on the instruction and objects, or if they also rely on the identical physical kitchen setup and object locations.
>
> The tasks considered in-distribution (ID) are considered ID because they contain objects and instructions inside the Bridgev2 dataset. As you pointed out, they also rely on our toy kitchen environment, which also belongs to the Bridgev2 dataset. We attempted to keep the camera position in a similar position to minimize distribution shift between the training data and the real-world, thereby helping to isolate the performance of our policy.

---

> > ### Author Response · Authors · 2025-11-25
> >
> > We thank the reviewer again for their feedback and wish to make an additional comment. If the suggestion to 'ablate different VLM models for the purpose of task curation' refers to changing the VLM backbone in our robot policy (rather than the labeling VLM), we wish to highlight that we have addressed this in response to another reviewer and include it below:
> >
> > Specifically, to evaluate whether our methodology is VLM model-agnostic, we trained VLM2VLA-Q using Qwen-3-8B-IT [1], instead of Gemma-3-12B-IT, with identical training data and hyperparameters to our original model. As demonstrated in Fig. 9 (Appendix, Section A), **over the course of 200 additional robot experiments**, VLM2VLA-Q achieves competitive performance across all manipulation tasks, confirming that our 'actions as language' approach generalizes successfully to different VLM backbones. While VLM2VLA-Q exhibits slightly lower task success rates compared to our primary model, we note that model capacity (8B vs. 12B parameters) is a confounding factor. Importantly, Fig. 10 (Appendix, Section A) shows that VLM2VLA-Q demonstrates comparable planning capabilities, suggesting the performance gap stems primarily from action prediction rather than reasoning capabilities.
> >
> > [1] Yang, A., Li, A., Yang, B., Zhang, B., Hui, B., Zheng, B., ... & Qiu, Z. (2025). Qwen3 technical report. arXiv preprint arXiv:2505.09388.

---

### Official Review · Reviewer_89io · 2025-10-31

**Soundness:** 3
**Presentation:** 4
**Contribution:** 3
**Rating:** 6
**Confidence:** 4

**Summary:**

This paper presents the VLM2VLA pipeline, which turns VLMs into VLAs while maintaining their generalization performance for OOD settings. It addresses the critical issue of VLA pretraining degrading VLM features over time, due to the large domain shift between VLM training and action generation. The concept is based on addressing the distribution mismatch between actions and text, which typically undermines the knowledge gained from VLM pretraining.

The method involves modeling actions using words and then fine-tuning the VLM on that action word data. Since the VLM has encountered many instances of numbers in the context of science texts, etc., it already has a good understanding of them and can easily grasp the concept of action logic. Crucially, it's a VLA training pipeline that uses LoRA to maintain the VLM features. It also uses a hierarchical reasoning setup in combination with the text actions: high-level subtask -> mid-level motion planning -> low-level action. Results on VLM benchmarks and various real-world manipulation setups demonstrate that the proposed method is able to preserve VLM knowledge while serving as a good VLA with improved OOD performance.

**Strengths:**

- This is an overall very insightful paper with good experiments and results, which is helpful for the VLA community.
- It also tackles a critical issue in VLA pretraining, where the VLA training process destroys the VLM's foundational features
- The Action as Language Representation is a novel with good results but limited ablations
- Demonstrating that this method can work with VLA training solely using LoRA is a significant advantage for achieving efficient VLA adoption from VLMs.
- The OOD experiments and results are promising and demonstrate that the method effectively maintains VLM generalization.

**Weaknesses:**

- The current main weakness of the paper is the weak ablations of their key contributions: no ablation of how the 3-stage word actions impact the training, with baselines that only predict low-level actions, mid-level ones, etc
- Low-level action has translational DoF only, without any more complicated movements
- Very low inference speed, with several seconds for a single action prediction
- No VLM-centric ablations. It would have been interesting to study how different VLM backbones affect the performance of this pipeline

**Questions:**

- Can you add ablations for the hierarchical word action setup to study the impact of each hierarchy level on the performance? Especially since all are generated by Gemini and not humans. How important are good plans? Would be adoption work also with false traces? More insights here would strengthen the paper
- Additional tests with other VLMs would also be great to see how the pipeline performs across this axis
- Is LORA for all training stages a consequence of limited compute? How would it behave with smaller VLMs and full finetuning? Any insights on LORA rank and downstream performance?
- Does it scale to SIMPLER benchmark evaluation with 7 DoF? If so, what are the results?

---

> ### Author Response · Authors · 2025-11-20
>
> We thank the reviewer for their thoughtful review and positive assessment. We appreciate the recognition of our contributions, and now we will address the individual concerns below:
>
> > The current main weakness of the paper is the weak ablations of their key contributions: no ablation of how the 3-stage word actions impact the training, with baselines that only predict low-level actions, mid-level ones, etc.
>
> We thank the reviewer for this critique. We are in the process of training models and conducting hardware experiments on an ablation which only outputs low-level actions, foregoing the reasoning (mid-level) steps. We will provide updated results once the experiments are complete.
>
> > Low-level action has translational DoF only, without any more complicated movements
>
> We acknowledge this limitation and explicitly discuss it in Section 5.1. This design choice was made to enable high-quality automatic labeling from Gemini-2.5. However, we emphasize that our framework naturally extends to rotational DoFs and may require stronger VLMs to perform this automatic relabeling. As noted in our limitations, we are actively exploring this direction. However, we wish to highlight that our core insight (representing actions as language to preserve VLM knowledge) remains valid regardless of DoF complexity.
>
> > Very low inference speed, with several seconds for a single action prediction
>
> We agree that inference latency is crucial and acknowledge that our current methodology is slow, relative to other state-of-the-art VLAs. However, we wish to highlight that inference speed optimization was not a main focus of this paper; rather, our work raises awareness to the importance of action representation in VLA pretraining, and reserve improvements in latency for future work (see Section 5.1).
>
> > No VLM-centric ablations. It would have been interesting to study how different VLM backbones affect the performance of this pipeline
>
> This is an insightful comment, and we are in the process of training models and running hardware experiments on a policy which uses a different VLM backbone than Gemma-3; we will update the manuscript once the results are finalized. However, we wish to state that while our method is architecture agnostic, we argue that providing an extensive ablation across many VLMs is computationally prohibitive given the time constraints of the rebuttal period, albeit we acknowledge that this would be an interesting ablation in future work.
>
> > Can you add ablations for the hierarchical word action setup to study the impact of each hierarchy level on the performance? Especially since all are generated by Gemini and not humans. How important are good plans? Would be adoption work also with false traces? More insights here would strengthen the paper
>
> As stated above, we are in the process of performing an ablation which foregoes all reasoning steps, only executing low-level actions in natural language. We will add another official comment once the results are finalized.
>
> > Additional tests with other VLMs would also be great to see how the pipeline performs across this axis
>
> As stated above, we are in the process of performing an ablation which replaces Gemma-3 with another VLM. We will add another official comment once the results are finalized.
>
> > Is LORA for all training stages a consequence of limited compute? How would it behave with smaller VLMs and full finetuning? Any insights on LORA rank and downstream performance?
>
> We used LoRA for VLA pretraining because we aimed to preserve the VLM backbone. Full-parameter fine-tuning often results in catastrophic forgetting, as evidenced by the VQA performance of VLAs shown in Table 1. While LoRA did enable memory savings, the main reason for using it was to prevent catastrophic forgetting. We did not conduct extensive LoRA rank ablations in our work but anticipate a larger rank would enable greater dexterity, at potentially the cost of less retained world knowledge. Likewise, we imagine smaller VLMs would require more aggressive LoRA fine-tuning (e.g., larger rank) for similar performance in manipulation.
>
> > Does it scale to SIMPLER benchmark evaluation with 7 DoF? If so, what are the results?
>
> We deliberately chose not to evaluate in SIMPLER. While simulation often serves as a proxy for downstream robot deployment, we believe that true out-of-distribution policy evaluation is only possible in the real-world. Moreover, extending to 7 DoFs runs into the same problem mentioned earlier: we currently do not have rotational DoFs in our training data, so cannot naively evaluate rotation in SIMPLER.

---

> > ### Author Response · Authors · 2025-11-25
> >
> > We again thank the reviewer for the above-mentioned concerns. We have completed ablation studies, including extensive hardware experiments amounting to **over 400 additional policy rollouts**. The updated manuscript now includes two additional models: 1) VLM2VLA-NR, a variant trained to output only language actions with no reasoning traces or subtask prediction, and 2) VLM2VLA-Q, which replaces the Gemma-3 backbone with Qwen-3-8B-IT [1]. Complete experimental details and results are provided in Appendix A. We summarize the key findings below:
> >
> > > Can you add ablations for the hierarchical word action setup to study the impact of each hierarchy level on the performance? Especially since all are generated by Gemini and not humans. How important are good plans? Would be adoption work also with false traces? More insights here would strengthen the paper
> >
> > The reviewer raises an important question regarding the contribution of each hierarchy level, particularly given that the reasoning traces are generated by Gemini, which may prove difficult to scale. To address this question, we trained VLM2VLA-NR, an ablation which forgoes subtask prediction and motion planning, directly predicting actions from visual observations and language instructions. As shown in Fig. 9 (Appendix, Section A), this variant exhibits degraded performance across all tasks, particularly in long-horizon and out-of-distribution scenarios. These results demonstrate that test-time reasoning is critical to our approach's success, validating the utility of our hierarchical action representation.
> >
> > We acknowledge that due to time constraints, we did not ablate the *quality* of the reasoning traces themselves, which is an important direction for future work. Understanding the tolerance of our approach to imperfect or noisy reasoning traces would provide valuable insights and could potentially enable our method to scale without requiring an external VLM to label robot datasets offline.
> >
> > > Additional tests with other VLMs would also be great to see how the pipeline performs across this axis
> >
> > To evaluate whether our methodology is model-agnostic, we trained VLM2VLA-Q using Qwen-3-8B-IT with identical training data and hyperparameters. As demonstrated in Fig. 9 (Appendix, Section A), VLM2VLA-Q achieves competitive performance across all manipulation tasks, confirming that our 'actions as language' approach generalizes successfully to different VLM backbones. While VLM2VLA-Q exhibits slightly lower task success rates compared to our primary model, we note that model capacity (8B vs. 12B parameters) is a confounding factor. Importantly, Fig. 10 (Appendix, Section A) shows that VLM2VLA-Q demonstrates comparable planning capabilities, suggesting the performance gap stems primarily from action prediction rather than reasoning capabilities.
> >
> > [1] Yang, A., Li, A., Yang, B., Zhang, B., Hui, B., Zheng, B., ... & Qiu, Z. (2025). Qwen3 technical report. arXiv preprint arXiv:2505.09388.

---

### Official Review · Reviewer_75Ka · 2025-11-01

**Soundness:** 3
**Presentation:** 3
**Contribution:** 3
**Rating:** 6
**Confidence:** 4

**Summary:**

This paper proposes VLM2VLA, a data-centric method for fine-tuning Vision-Language Models (VLMs) into Vision-Language-Action (VLA) models while minimizing catastrophic forgetting. The key idea is to represent robot actions as natural language phrases rather than discrete tokens or continuous vectors, allowing fine-tuning via LoRA without modifying the base VLM architecture.

**Strengths:**

1. Casting low-level actions as text tokens is conceptually elegant and well aligned with the VLM’s pretraining distribution, avoiding token-level mismatches common in prior VLAs.
2. Experiments show that multimodal reasoning and multilingual understanding are preserved, as in across diverse VQA benchmarks and real-world robotic tasks.
3. The proposed pipeline of using LoRA fine-tuning only avoids costly co-training or architecture modifications.

**Weaknesses:**

1. The comparison with baselines on robotics tasks are limited. MolmoAct and π0.5 are mentioned in Table 1 but not reported in robot experiments. Other VLA works that address the catastrophic forgetting problems[2] or unifying the architectures[1] are not compared.
2. Robot experiments are relatively simple and there are no simulation experiments, which could have provided scalable quantitative metrics.
3. While compared with an action expert ablation with discrete tokens, there is no comparison with flow-matching or diffusion-based action experts such as π0.5 and MolmoAct, which provides state-of-the-art continuous control.
4. The proposed three-stage hierarchy (subtask → motion plan → action chunk) is promising but conceptually overlaps with system-1/system-2 reasoning paradigms (planning vs. acting). A comparison with a 2 system VLA is needed to validate the proposed method, where a system2 is responsible for planning and translating and a system1 is responsible for action generation.

[1] Lin, Fanqi, et al. "OneTwoVLA: A Unified Vision-Language-Action Model with Adaptive Reasoning." arXiv preprint arXiv:2505.11917 (2025).

[2] Huang, Huang, et al. "Otter: A vision-language-action model with text-aware visual feature extraction." arXiv preprint arXiv:2503.03734 (2025).

**Questions:**

1. What’s the X axis of figure 6 and does the images of fish and country flag correspond to the plot?
2. Why only are translational dofs considered?

---

> ### Author Response · Authors · 2025-11-19
>
> We thank the reviewer for their kind assessment of our work and for recognizing our experimental validation. Next, we address the individual concerns raised by the reviewer below.
> > The comparison with baselines on robotics tasks are limited. MolmoAct and π0.5 are mentioned in Table 1 but not reported in robot experiments. Other VLA works that address the catastrophic forgetting problems[2] or unifying the architectures[1] are not compared.
> We appreciate the reviewer’s suggestion to compare to additional recent VLA works, specifically, OneTwoVLA [1] and OTTER [2], and clarify why we don’t experiment with them in our work.
>
> OneTwoVLA [1] and OTTER [2] address different conceptual problems than ours. While [1] is a reasoning VLA like our work, they focus on adaptive reasoning and co-train on embodied VQA tasks to improve generalization; we deliberately avoid co-training in this work. OTTER highlights the importance of vision-text alignment in preserving the vision encoder’s latent pretraining knowledge. While OTTER is philosophically more related to our work, they do not show this alignment strategy retains VQA performance. Moreover, they train their policy backbone from scratch, whereas we use a pretrained VLM to create our VLA.
>
> Lastly, we include a VQA analysis of MolmoAct and $\pi_{0.5}$ because they are popular, state-of-the-art, open-source VLAs which co-train on VQA datasets. The decision to include these analyses was to quantify their VQA abilities, which is of independent interest to the robot learning community, and show that co-training may not be sufficient in preserving the VLM knowledge. We do not compare our method to these models in the hardware experiments because they both co-train on multimodal web datasets, thereby not serving as a comparable baseline.
> > Robot experiments are relatively simple and there are no simulation experiments, which could have provided scalable quantitative metrics.
>
> We thank the reviewer for this comment and acknowledge that we did not perform any simulation experiments. While simulation may provide insight into real-world performance, we believe that true generalization can only be tested through real-world evaluation. Out-of-distribution (OOD) performance is most apparent in real hardware experiments, where the model must contend with real variations that cannot be easily controlled for in simulation. While we do acknowledge that our work only focused on pick-and-place experiments, we wish to highlight that these are common for VLA policy evaluation and are within the scope of our VLA pretraining dataset.
> > While compared with an action expert ablation with discrete tokens, there is no comparison with flow-matching or diffusion-based action experts such as π0.5 and MolmoAct, which provides state-of-the-art continuous control.
>
> Similar to our comments above, we do not compare to $\pi_{0.5}$ nor MolmoAct in our hardware experiments because both co-train on multimodal VQA datasets, thereby not serving as comparable baselines.
> > The proposed three-stage hierarchy (subtask → motion plan → action chunk) is promising but conceptually overlaps with system-1/system-2 reasoning paradigms (planning vs. acting). A comparison with a 2 system VLA is needed to validate the proposed method, where a system2 is responsible for planning and translating and a system1 is responsible for action generation.
>
> Our approach is similar in spirit to the system-1/system-2 approaches. Indeed, we do not compare to models which utilize distinct instantiations of the system-1 and system-2 processes for a similar reason to why we did not compare to $\pi_{0.5}$ nor MolmoAct; they offer a fundamentally different approach to our method. In this work, our core claim is that the action as language representation helps enable VLA generalization to novel tasks by better leveraging the VLM backbone. Comparing to methods which separate the two is orthogonal to what our work is trying to achieve: knowledge transfer from the VLM in downstream robotic tasks.
>
> > What’s the X axis of figure 6 and does the images of fish and country flag correspond to the plot?
>
> We apologize for the confusion. The x-axis shows different OOD evaluation conditions: the country flags represent the three languages tested in the multilingual task (Spanish, Mandarin, Hindi), and the Ash Ketchum image represents the semantic reasoning task requiring pop-culture knowledge. We will clarify the axis labels and captions in updated PDF.
>
> > Why only are translational dofs considered?
>
> We only considered translational DOFs out of practicality, as this was a design choice enabling high-quality automatic labeling from Gemini-2.5 (Section 3.2). As VLMs improve at spatial reasoning, our framework naturally extends to rotational DoFs and we explicitly discuss this limitation in Section 5.1; we are exploring this avenue in future work.

---

### Official Review · Reviewer_9qWQ · 2025-11-01

**Soundness:** 3
**Presentation:** 3
**Contribution:** 2
**Rating:** 2
**Confidence:** 4

**Summary:**

The paper focus on a reasonable issue, finetuning VLM to VLA will cause the VLM lost its memory for QA but overfit to robot control. The paper proposes to solve this issue by convert the low level action to natural language format, then use lora finetuning.

**Strengths:**

1. The scope of this paper makes sense, which i agree that fine-tuning the VLM to VLA wil let the VLA lost its QA reasoning ability and overfit to specific control actions prediction.

2. The paper is well organized and easy to follow.

**Weaknesses:**

1. Even the paper throws a meaningful problem, which is how to fine-tune a vlm to vla without degrade its ablity for reasoning, but the paper's solution is not cool enough, since just format low level actions as language then train it using formal vlm training, this is not novel and used by many papers like [1][2]

2. Not sure what is the paper's main contribution, seems Table 1 want to claim VLM2VLA performs better in series VQA task, but, do the VQA person will use VLA for VQA task, or Robotics ppl care these VQA task? So not sure if evaluate on these VQA task really make sense if this work focus on VLA.


[1] Niu, D., Sharma, Y., Biamby, G., Quenum, J., Bai, Y., Shi, B., ... & Herzig, R. (2024). Llarva: Vision-action instruction tuning enhances robot learning.
[2] Zitkovich, B., Yu, T., Xu, S., Xu, P., Xiao, T., Xia, F., ... & Han, K. (2023, December). Rt-2: Vision-language-action models transfer web knowledge to robotic control. In Conference on Robot Learning (pp. 2165-2183). PMLR.

**Questions:**

See Weakness for big concern. Besides those:
1. Seems in section 4.2, it shows some results on robotic control task, but there is some conerns:
a. it deos not compare with a strong baseline, like pi0
b. the task is too simple, mainly pick and place, which i think should focus more on long horizon task, which might better show the method can keep more reasoning or planning ability of the pretrianed VLM.

---

> ### Author Response · Authors · 2025-11-19
>
> We thank the reviewer for their feedback and for acknowledging that our paper addresses a meaningful problem in robot learning with clear presentation. Next, we address the individual concerns raised by the reviewer below.
>
> > Even the paper throws a meaningful problem, which is how to fine-tune a vlm to vla without degrade its ablity for reasoning, but the paper's solution is not cool enough, since just format low level actions as language then train it using formal vlm training, this is not novel and used by many papers like [1][2]
>
> We argue that our contribution is novel and distinct from the cited works: While LLARVA [1] uses numerical text for actions and LoRA fine-tuning, their focus is on visual trajectory traces and cross-embodiment generalization. Critically, they do not investigate catastrophic forgetting, compare action representations, nor evaluate VQA abilities of their LLM and vision encoder architecture pre- and post-training. Our work demonstrates that the choice of action representation impacts downstream generalization in robotic tasks, a question LLARVA does not address (see the semantic generalization tasks considered in Section 4.2).
>
> With respect to RT-2 [2], that work prevents catastrophic forgetting through co-training with massive web-scale VQA datasets. This is computationally expensive and requires finding the correct data mixture for effective performance. In contrast, we show that co-training can be eliminated entirely by resolving the distribution mismatch at the data level through our action representation. Moreover, RT-2 uses least-likely token assignment for actions, which is fundamentally different from our approach in that we do not modify the existing VLM vocabulary to accommodate action tokens.
>
> In summary, our key contributions are as follows:
> * We provide a demonstration showing that representing actions as language, as opposed to arbitrary token assignment, enables effective LoRA-only training without catastrophic forgetting (Table 1, VLM2VLA-AT ablation).
> * We empirically validate our preserved VLM knowledge enables out-of-distribution (OOD) task generalization requiring semantic reasoning (Section 4.2, Fig. 5-7).
>
> The insight that the choice of action representation determines whether or not parameter-efficient fine-tuning suffices for downstream robotic control and generalization to new tasks is non-obvious, and we believe the answer provided in this work yields new insights otherwise absent in the field.
>
> > do VQA person will use VLA for VQA task, or Robotics ppl care these VQA task?
>
> This question seems to fundamentally misunderstand why we evaluate our VLA on VQA tasks at all.
>
> We are not proposing VLAs as replacements for VLMs on VQA tasks. Rather, VQA benchmarks serve as tests to quantify whether the VLA has retained the semantic and perceptual understanding from its VLM pretraining. Why does this matter for robotics? The promise of using VLMs in VLAs is precisely in leveraging the latent world knowledge of the VLM, and we show this is what enables generalization to novel tasks, e.g., multilingual instruction following and zero-shot semantic reasoning.
>
> As we demonstrate empirically, VLAs that catastrophically forget (OpenVLA, ECoT) fail on these OOD manipulation tasks, while VLM2VLA succeeds. The VQA results in Section 4.1 help predict and explain this gap in manipulation performance showcased in Section 4.2.
>
> > Seems in section 4.2, it shows some results on robotic control task, but there is some conerns: a. it deos not compare with a strong baseline, like pi0 b. the task is too simple, mainly pick and place, which i think should focus more on long horizon task, which might better show the method can keep more reasoning or planning ability of the pretrianed VLM.
>
> We compare against OpenVLA and ECoT, which are state-of-the-art autoregressive VLAs that can run out-of-the-box on our hardware without additional fine-tuning; in particular, we compare against ECoT because its a reasoning VLA, like ours, and is trained on an identical dataset (Bridgev2). We do not compare manipulation performance against $\pi_{0.5}$ because they co-train on multimodal webdata, thereby not serving as a comparable baseline. However, we do perform a VQA analysis for $\pi_{0.5}$ in Section 4.1 to show that co-training alone may not be sufficient for knowledge preservation.
>
> Additionally, we argue that we do evaluate a long-horizon task: "Put the Eggplant In the Pan, Then Lift the Fish" requires multi-step planning with two distinct tasks. As shown in Fig. 5, VLM2VLA achieves the best success rates, in comparison to baselines, and Figure 6 demonstrates strong planning ability (with specific details in the Appendix). Also, our OOD tasks qualitatively differ from standard pick-and-place tasks in that they probe the latent 'reasoning or planning ability of the pretrained VLM' that the reviewer is interested in.

---

### Author Response · Authors · 2025-12-01
**Dear Area Chair and Reviewers**

We would like to thank all reviewers for their feedback thus far. In light of the area chair reassignment, we would like to take this opportunity to briefly summarize the status and updates of our paper.

**Individual Reviewer Responses**

- **Reviewer 75Ka** provided a positive assessment of our work with an initial score of 6, while raising concerns about limited baseline comparisons and the simplicity of our hardware experiments. We addressed these concerns by explaining our rationale for baseline selection (avoiding co-trained VLAs that solve the problem of catastrophic forgetting with an orthogonal approach), and justifying our focus on real-world experiments over simulation. The updated manuscript also includes clarifications to figures as requested by the reviewer.

- **Reviewer 89io** assigned a score of 6 and provided positive feedback while requesting ablation studies of our hierarchical action representation, VLM backbone selection, and LoRA design decisions. In response, we completed over 400 additional hardware experiments, adding two new ablation policies: VLM2VLA-NR, a policy without reasoning traces, and VLM2VLA-Q, a policy with a Qwen-3 backbone instead of Gemma-3. These results, included in the Appendix, validate the importance of hierarchical reasoning and demonstrate our method’s generalizability across VLM backbones.

- **Reviewer HTUd** gave a score of 8 and provided positive feedback while requesting additional baseline comparisons on real-world tasks, VQA evaluations for baseline VLM backbones, and clarifications on experimental design. We addressed these concerns by adding VQA results for PaliGemma and Molmo, the VLM backbones for $\pi_{0.5}$ and MolmoAct, to Table 1 of our manuscript. Moreover, we explained why direct hardware comparisons with these policies would not be appropriate due to their co-training procedures, and clarified our task definitions and experimental setup.

- **Reviewer 9qWQ** originally assigned a score of 2, raising concerns about novelty, experimental design, and the relevance of our VQA evaluation. We have addressed these concerns by clarifying how our contributions differ fundamentally from prior work, justifying our selection of baseline policies, and explaining the purpose of our VQA analysis. There has been no further discussion with the reviewer following our response.

**Summary**

All reviewers raised valid concerns and offered constructive criticism which we have used to improve our paper. During the rebuttal period, **we conducted over 400 additional real-world hardware experiments and provided detailed comments to each concern**. In particular:

- We justified our choice of baselines and real-world hardware experiments in response to Reviewer 75Ka

- We completed two ablation studies in response to Reviewer 89io, validating our hierarchical action representation and model-agnostic VLA training procedure.

- We performed two additional VQA analyses and provided comprehensive experimental clarifications for Reviewer HTUd

- We addressed concerns about novelty and experimental design raised by Reviewer 9qWQ.

We believe these updates have strengthened the paper and have addressed all major concerns raised during the review process.

Sincerely,

The Authors

---

### Meta-Review · Area_Chair_tCvD · 2026-01-03

**Summary:**

My recommendation for this paper is Accept. Most reviewers except one reviewer are positive about the paper. The main concern of the negative reviewer (9qWQ) is around novelty, contribution, missing baselines, and simple experimental tasks. Authors have clarified the main novelty and contributions of the paper. Other reviewers also highlighted this point, and I stand with this viewpoint. However, I don't recommend this paper for the award -- the paper is missing comparison to orthogonal yet important baselines in the community, the paper is missing evaluation in simulated tasks which can be helpful for future researchers, and have some weaknesses in its limited setup (translational dof) and slow inference time. I would like to recommend authors to include additional results in the final version of the paper if available as it would further strengthen the paper.

**Reviewer Concerns:**

Concerns resolved by the rebuttal
- Novelty and main contributions (Reviewer 9qWQ): Authors clarified the contributions
- Tasks are simple (Reviewer 9qWQ and 75Ka): Authors clarified the tasks are not super simple:
- No comparison to System-1 & System-2 approaches (Reviewer 75Ka): I agree with authors in that this comparison is out of scope
- Ablation studies (Reviewer 89io): Authors provided additional ablation studies
- Translational DoF only (Reviewer 89io, Reviewer 75Ka): Authors acknowledged the limitations
- Very low inference speed (Reviewer 89io): Authors acknowledged the limitations.
- VQA performance with VLM models from pi0 and MolmoAct (Reviewer HTUd): Authors provided additional experiments

Outstanding concerns
- Missing baselines (pi0.5, MolmoAct): Several reviewers pointed out the paper is missing these baselines. Authors claimed that they are orthogonal and therefore would not include the results in the paper. I disagree with this point. Including such results would still make the paper more *complete* and make the results easy-to-understand for the readers in the broad context. It's still possible to include such orthogonal results in the main figures/tables with a clear note
- Missing diffusion/flow-based baselines: Reviewer 75Ka specifically pointed out missing baselines that use diffusion or flow based models. This is different from the baselines that use co-training. This can be still implemented without such co-training.
- Simulation experiments: Reviewer 75Ka and 89io requested simulation experiments, authors did not provide additional experiments. While I agree that real-world robot experiments are informative, simulation experiments are still helpful as they serve as a reproducible, comparable baselines for future research.

**Reviewer Scores:**

- Reviewer 9qWQ (score 2): I expect this reviewer to increase score to borderline score (4 or 6, not completely sure)
- Reviewer 75Ka (score 6): I expect this reviewer to maintain their score due to missing comparison to Pi0.5 or MolmoAct
- Reviewer 89io (score 6): I expect this reviewer to maintain their score due to missing simulated experiments
- Reviewer HTUd (score 8): I expect this reviewer to maintain their score, and not to strongly champion the paper due to missing comparison to Pi0.5 and MolmoAct.

---

### Decision · Program_Chairs · 2026-01-26

Accept (Poster)